# Constrained Discrete Black-Box Optimization using Mixed-Integer Programming

## Abstract

Discrete black-box optimization problems are challenging for model-based optimization (MBO) algorithms, such as Bayesian optimization, due to the size of the search space and the need to satisfy combinatorial constraints. In particular, these methods require repeatedly solving a complex discrete global optimization problem in the inner loop, where popular heuristic inner-loop solvers introduce approximations and are difficult to adapt to combinatorial constraints. In response, we propose NN+MILP, a general discrete MBO framework using piecewise-linear neural networks as surrogate models and mixed-integer linear programming (MILP) to optimize the acquisition function. MILP provides optimality guarantees and a versatile declarative language for domain-specific constraints. We test our approach on a range of unconstrained and constrained problems, including DNA binding and the NAS-Bench-101 neural architecture search benchmark. NN+MILP surpasses or matches the performance of algorithms tailored to the domain at hand, with global optimization of the acquisition problem running in a few minutes using only standard software packages and hardware.

## 1 Introduction

The problem of optimizing an expensive black-box function $f : \Omega \mapsto \mathbb{R}$ over a discrete, constrained domain arises in numerous application domains, e.g. neural architecture search (Zoph & Le, 2017), program synthesis (Summers, 1977; Biermann, 1978), small-molecule design (Elton et al., 2019), and protein design (Yang et al., 2019). In such resource-constrained settings, it is desirable to develop algorithms that exploit known combinatorial structure in $\Omega$ to search the space more efficiently.

Model-based Black-box Optimization (MBO), a popular paradigm that includes Bayesian Optimization as a special case, iteratively refines a function approximator $\hat{f}(x) \approx f(x)$ and selects new points to query by optimizing an *acquisition function* $a(x)$ derived from a point estimate or posterior distribution over $\hat{f}$ (Section 2.1). This *inner-loop optimization* problem is assumed to be easier than the original, since, for example, $a(x)$ is less expensive to query than $f(x)$ or provides "white-box" properties such as gradients.

There is a vast literature addressing the challenges of applying MBO in practice. We focus on two of these: first, optimizing $a(x)$ may itself be a computationally-difficult optimization problem; second, in many applications, practitioners are confronted by additional constraints on $x$. For example, in neural architecture search, $x$ might represent a computation graph that must be both connected and acyclic. Due to the difficulty in optimizing the acquisition function over a combinatorial domain, most approaches resort to heuristic inner-loop solvers, which often need to be specialized to the problem at hand to ensure feasibility, e.g. evolutionary solvers with custom mutation operators.

To address the challenge of inner-loop optimization, we introduce a new MBO framework, *NN+MILP*, that *exactly* solves the acquisition problem using mixed-integer linear programming (MILP). Crucially, by framing the inner-loop optimization as an MILP, our approach can flexibly incorporate a wide variety of logical, combinatorial, and polyhedral constraints, which need only be provided in a *declarative* sense. Using MILP in the inner loop does restrict the functional form of $\hat{f}$ (or the acquisition function based on it), but it supports any piecewise linear function. In particular, we employ the class of neural network (NN) approximators with ReLU activation functions due to their scalability and accuracy in practice, and because we can draw on recent work improving the

performance of MILP for optimizing such NNs with respect to their inputs (Anderson et al., 2020). For us, MILP is practical to use in the inner loop because the dimensionality of typical black-box optimization problems is orders of magnitude smaller than those usually considered by MILP solvers. Our contributions are as follows:

- We introduce *NN+MILP*, an MBO framework for discrete black-box problems with NN surrogates and exact optimality guarantees for solving the acquisition problem.
- We observe in our experiments that the runtime of MILP is practical for use with black-box problems of real-world scale, often solving the inner acquisition problem in seconds using standard packages and hardware. Moreover, MILP provides a simple declarative language for problem-specific constraints.
- We show that NN+MILP matches or surpasses the performance of strong MBO baselines based on problem-specific evolutionary algorithms on a wide range of synthetic and real-world discrete black-box problems.
- We use the NAS-Bench-101 neural architecture search benchmark as a case study, presenting a novel MILP formulation of its graph-structured domain.

## 2 BACKGROUND AND RELATED WORK

### 2.1 MODEL-BASED BLACK-BOX OPTIMIZATION

Model-based Black-box Optimization (MBO) is a broad family of methods that includes Bayesian optimization as a special case (Mockus et al., 1978; Jones et al., 1998; Hutter et al., 2011; Snoek et al., 2012; Shahriari et al., 2015). As depicted in Algorithm 1, the method proposes $x_t$ at iteration $t$ using three steps. First, the user performs inference over a *surrogate model* $\hat{f}$ to approximate $f$ using the data previously collected from the black-box function. Here, `fit()` may return a point estimate for $\hat{f}$, a posterior distribution over $\hat{f}$, or a posterior predictive distribution. Next, an *acquisition function* $a(x)$ based on $\hat{f}(x)$ is posed that quantifies the quality of new points to query. Finally, $x_t$ is selected as the best point found by solving the *acquisition problem*, where an *inner-loop solver* (approximately) optimizes $a(x)$. The acquisition problem is typically designed such that it is more approachable than directly solving the original problem. For example, $a(x)$ may be orders of magnitude less expensive to evaluate or have a tractable functional form. Practitioners can encode prior knowledge about the structure of $f$ via a choice of inductive bias for $\hat{f}$, e.g. a suitable Gaussian Process kernel or neural-network architecture.

Bayesian optimization performs Bayesian inference over $\hat{f}$ and employs an acquisition function that accounts for uncertainty in $\hat{f}$. Doing so provides principled mechanisms for balancing exploration and exploitation (Mockus et al., 1978; Srinivas et al., 2010) and is particularly important in early rounds of optimization when models are fit on limited data. We refer to our method as an instance of MBO, not Bayesian optimization, because it does not assume formal Bayesian inference for $\hat{f}$. Gaussian processes (GPs) are often used for $\hat{f}$ in Bayesian optimization, since they provide closed-form posterior inference, naturally adjust their expressivity as the dataset grows, and users can inject domain knowledge via a choice of kernel (Rasmussen & Williams, 2006; Oh et al., 2019). On the other hand, neural networks provide a practical alternative (Snoek et al., 2015; Hernández-Lobato et al., 2017), since they often scale more gracefully, either computationally or statistically, to large datasets or high-dimensional domains.

### 2.2 SOLVING THE DISCRETE MBO ACQUISITION PROBLEM

In general, the inner-loop problem is itself a non-trivial global optimization problem. Prior work on discrete MBO has mainly employed local search solvers, such as evolutionary search, with limited guarantees (Hutter et al., 2011; Müller, 2016; Oh et al., 2019; Kandasamy et al., 2020). A key advantage of such solvers is that they treat $a(x)$ as a black box, which provides practitioners with freedom when designing application-specific surrogate models. On the other hand, particular choices of surrogate model and acquisition function lead to acquisition problems that can be (approximately) solved using specialized combinatorial solvers (Baptista & Poloczek, 2018; Deshwal et al., 2020), mixed-integer nonlinear programming (MINLP) solvers (Costa & Nannicini, 2018; Kim & Boukouvala, 2020), or continuous optimization solvers (Bliek et al., 2021).

| **Algorithm 1** MBO | **Algorithm 2** NN+MILP |
|---|---|
| **Input:** hypothesis class $\mathcal{F}$, budget $N$, initial dataset $\mathcal{D}_n = \{x_i, f(x_i)\}_{i=1}^n$, optimization domain $\Omega$ | **Input:** hypothesis class $\mathcal{F}$, budget $N$, initial dataset $\mathcal{D}_n = \{x_i, f(x_i)\}_{i=1}^n$, MILP domain formulation $\mathcal{M}_\Omega$ |
| **for** $t = n+1$ to $t = N$ **do** | **for** $t = n+1$ to $t = N$ **do** |
| $\quad P(\hat{f}_t) \leftarrow \texttt{fit}(\mathcal{F}, \mathcal{D}_{t-1})$ | $\quad \hat{f}_t \leftarrow \texttt{fit}(\mathcal{F}, \mathcal{D}_{t-1})$ $\qquad\qquad$ (3.2) |
| $\quad a(x) \leftarrow \texttt{get\_acquis\_func}(P(\hat{f}_t))$ | $\quad \mathcal{M}_t \leftarrow \texttt{build\_milp}(\hat{f}_t, \mathcal{M}_\Omega, \mathcal{D}_{t-1})$ $\quad$ (3.3) |
| $\quad x_t \leftarrow \texttt{inner\_solver}(a(x), \Omega)$ | $\quad x_t \leftarrow \texttt{optimize}(\mathcal{M}_t)$ (generic MILP solver) |
| $\quad \mathcal{D}_t \leftarrow \mathcal{D}_{t-1} \cup \{x_t, f(x_t)\}$ | $\quad \mathcal{D}_t \leftarrow \mathcal{D}_{t-1} \cup \{x_t, f(x_t)\}$ |
| **end for** | **end for** |
| **return** $\arg\max_{(x_t, y_t) \in \mathcal{D}_N} y_t$ | **return** $\arg\max_{(x_t, y_t) \in \mathcal{D}_N} y_t$ |

Therefore, practitioners must decide between either introducing difficult-to-analyze approximations due to inexact heuristic solvers or using tractable surrogate models that may be mis-specified for the application domain. This serves as a key motivation for our work: we seek to enable practitioners to employ broad families of surrogate models and exactly solve the acquisition problem with reasonable computational overhead in practice.

## 2.3 Constrained MBO

In many applications, $x$ is subject to non-trivial structural constraints. Prior work has largely focused on the case where determining whether $x$ is feasible requires evaluating an expensive, perhaps noisy, black-box function $h(x)$ with cost comparable to $f(x)$ (Schonlau et al., 1998; Gelbart et al., 2014; Hernández-Lobato et al., 2016; Ariafar et al., 2019; Letham et al., 2019). Here, standard acquisition functions can be extended to account for an additional classifier $\hat{h}(x)$ trained to predict $h(x)$.

Problems with inexpensive white-box $h(x)$ can be tackled using these approaches for black-box constraints, but doing so may lead to slower optimization and may query $f(x)$ at invalid $x$, which can be unsafe when performing physical experiments (Berkenkamp et al., 2016). Instead, the inner-loop solver can be modified directly to guarantee feasibility, e.g., by using rejection sampling (Shi et al., 2020; Kandasamy et al., 2020). If using local search algorithms, the solver would need to be customized for each family of constraints, a task usually left to the user. Prior work employing MINLP solvers addresses white-box constraints either by adding a penalty for constraint violation (Costa & Nannicini, 2018) or in small-scale settings (Kim & Boukouvala, 2020).

## 2.4 Mixed Integer Linear Programming

Mixed Integer Linear Programming (MILP) seeks to maximize a linear function over a set of decision variables, some of which may be integral, subject to linear inequality constraints. Decades of development have allowed MILP to have a significant impact in a wide range of applications due to its better-than-expected computational performance (Jünger et al., 2010). Indeed, while MILP problems are computationally hard (NP-complete), they are routinely solved (to global or near-global optimality) in production environments thanks to state-of-the-art solvers that nearly double their machine-independent performance every year (Achterberg & Wunderling, 2013; Bixby, 2012).

A notable aspect of MILP is that it provides a simple yet extremely versatile declarative language for white-box constraints. It is well known that linear inequalities over integer variables can be used to easily build *pure-integer* formulations for logical constraints and combinatorial optimization problems (Williams, 2013; Schrijver, 2003; Wolsey & Nemhauser, 1999). In addition, using both integer and continuous variables leads to *mixed-integer* formulations that can combine polyhedral and logical constraints (Jeroslow, 1989; Pochet & Wolsey, 2006; Vielma, 2015).

Particularly interesting to our proposed approach are MILP formulations for piecewise-linear functions (Huchette & Vielma, 2019; Vielma et al., 2010). Specifically, our work leverages MILP formulations for trained neural networks with piecewise-linear activation functions such as ReLUs (Anderson et al., 2020). Optimizing over trained ReLU networks with MILP has been done in contexts such as neural network verification (Cheng et al., 2017; Lomuscio & Maganti, 2017; Tjeng et al., 2019), reinforcement learning (Ryu et al., 2020; Delarue et al., 2020), and analysis and exact com-

pression of neural networks (Serra et al., 2018; 2021). In particular, MILP has also been used to optimize ReLU network surrogates of simulation-based constraints (Grimstad & Andersson, 2019), although their approach optimizes a single surrogate model once, unlike in ours.

## 3 MILP FOR MBO

We propose the *NN+MILP* framework (Algorithm 2), which uses neural network surrogate models and solves the acquisition problem using MILP at every step. This provides practitioners with the flexibility to use a wide variety of models and leverage MILP's versatile declarative language to incorporate constraints. This section describes various design choices to make the approach practical.

### 3.1 PROBLEM SETTING

Our goal is to find:

$$x^* = \arg \max_{x \in \Omega} f(x), \tag{1}$$

where $f : \Omega \mapsto \mathbb{R}$ is an expensive, noiseless black-box reward function and $\Omega \subseteq \Omega_1 \times \ldots \times \Omega_n$ is a domain on $n$ decision variables. We assume $\Omega$ can be described by an inexpensive function $h_\Omega(x)$ indicating whether $x$ is in $\Omega$. Algorithms are allowed a fixed budget of $N$ sequential queries to $f$. $\mathcal{X}_t := \{x_i\}_{i=1}^t$ refers to the set of sampled points by iteration $t$, and $\mathcal{D}_t := \{x_i, y_i = f(x_i)\}_{i=1}^t$ includes corresponding rewards. An algorithm's performance is measured as the best reward in $\mathcal{D}_N$. Since $f$ is noiseless, it is advantageous for algorithms to avoid repeated evaluations of the same $x$.

We choose to focus on finite discrete sets $\Omega$ as we believe this is the area where MILP can provide the greatest benefit. As noted in Section 2.4, there are many well-studied formulation techniques for $\Omega$ with combinatorial structure, such as directed graphs. More generally, such sets have a polynomially-sized MILP formulation whenever $h_\Omega(x)$ can be evaluated in polynomial time (e.g., Yannakakis (1991)). Continuous and mixed-integer domains could be incorporated in our approach with some modifications (Section 6), although they are outside the scope of this paper.

### 3.2 SURROGATE MODEL AND ACQUISITION FUNCTION

For surrogate model $\hat{f}$, we allow any feedforward neural network with piecewise-linear activation functions, as they can be represented by MILP (Section 3.3). Though we focus on fully-connected networks using ReLUs, we note that a wide range of architectures (e.g., those including convolutional and max-pooling layers) are piecewise-linear and may place suitable inductive bias on $\hat{f}$.

Our experiments employ a simple acquisition function loosely motivated by Thompson sampling (Thompson, 1933; Hernández-Lobato et al., 2017; Kandasamy et al., 2018). At each optimization step, we train a new regressor $\hat{f}(x)$ from scratch and set $a(x) = \hat{f}(x)$. This relies on stochastic gradient descent training and random parameter initialization to increase the variability in surrogate models across iterations (Lakshminarayanan et al., 2017). We discuss extensions to alternative acquisition functions in Section 6. We use a flattened one-hot encoding of $x$ for the input layer, and train each network $\hat{f}_t \in \mathcal{F}$ on $\mathcal{D}_{t-1}$ using $\ell_2$ loss. Finally, we re-scale the observed rewards in $\mathcal{D}_{t-1}$ before training models, to aid both in training and optimization. Poorly-scaled data may result in slower performance or small inaccuracies in MILP solvers (Miltenberger et al., 2018).

### 3.3 MILP FORMULATION OF THE ACQUISITION PROBLEM

The inner-loop solver then seeks to find

$$x_t = \arg \max_{x \in \Omega \setminus \mathcal{X}_{t-1}} \hat{f}_t(x), \tag{2}$$

where $\Omega$ is the feasible set for (1) and $\mathcal{X}_{t-1}$ is the set of points where the noiseless $f(x)$ has been queried already. The MILP formulation of (2) is denoted by $\mathcal{M}_t$ and has the following three components:

**Domain** We use a one-hot encoding of decision variables $x$ (unless they are already binary), defining the binary decision vector $z$ with $z_{ij} \equiv \mathbb{I}\{x_i = j\}$ for $i \in [n], j \in \Omega_i$, and subject to linear

constraints $\sum_{j \in \Omega_i} z_{ij} = 1 \; \forall i$. Integer domains with small range may be one-hot encoded; see Appendix D for a comparison between integer and one-hot encodings. Additional constraints due to $\Omega$ are added as necessary, with form dependent on the application at hand. We assume that these are MILP-representable, which as noted in Section 2.4 could include a wide range of combinatorial, logical, and polyhedral constraints. We use $\mathcal{M}_\Omega$ to denote the domain formulation itself.

**No-good Constraints** A *no-good constraint* is one that eliminates one or more undesirable solutions from the domain. Here, we leverage the binary nature of $z$ to *exactly* eliminate the set $\mathcal{X}_{t-1}$ from $\mathcal{M}_t$. For illustrative purposes, consider a single point $\bar{x} \in \Omega$ we wish to exclude from the acquisition problem's domain, and let $\bar{z}$ denote its one-hot encoding (or $\bar{x}$ itself if the problem is binary). Then the constraint:

$$\sum_{i,j \,:\, \bar{z}_{ij}=0} z_{ij} + \sum_{i,j \,:\, \bar{z}_{ij}=1} (1 - z_{ij}) \geq 1 \tag{3}$$

enforces that any feasible $z$ has a Hamming distance of at least 1 from $\bar{z}$. As $z$ are binary, this effectively eliminates just the single point $\bar{z}$ from the feasible region. We therefore formulate $\Omega \setminus \mathcal{X}_{t-1}$ by including one such constraint for each $\bar{x} \in \mathcal{X}_{t-1}$. Note that the right-hand side can be tightened to 2 for one-hot encodings, and these compact no-good constraints do not extend naturally to continuous $x$ (Section 6).

**Neural Network** We formulate the neural network by introducing auxiliary decision variables encoding the activation of each neuron for a given $z$. We present here the formulation for a single ReLU, commonly used throughout the literature (Section 2.4), while noting that the full formulation is obtained by combining all ReLU formulations and matching their input and output variables according to the structure of the network. The overall MILP objective is the activation corresponding to the regressor's output neuron.

A ReLU neuron with vector input $x$ and scalar output $y$ has the piecewise-linear form $y = \max(0, w^\top x + b)$, where $w$ and $b$ are its weights and bias respectively. At optimization time, $w$ and $b$ are fixed, while $x$ and $y$ are represented by decision variables (also used as the inputs and outputs of other ReLUs according to the feedforward structure). To handle the non-linearity, we add a binary decision variable $\alpha$ that indicates whether the ReLU is active or not. We then write the following set of constraints to enforce that $y = 0$ when $\alpha = 0$ and $y = w^\top x + b$ when $\alpha = 1$:

$$0 \leq y \leq M\alpha \tag{4}$$

$$w^\top x + b \leq y \leq w^\top x + b + M(1 - \alpha) \tag{5}$$

where $M$ is a sufficiently large fixed value, such as an upper bound on the range of $y$. As $w$ and $b$ are fixed, values for $M$ can be computed in advance of the optimization, e.g. by propagating bounds from $\Omega$. Our experiments use a more advanced method to compute $M$, detailed in Appendix A.

### 3.4 Optimality Guarantees for MILP

The full acquisition problem formulation, denoted by $\mathcal{M}_t$, is passed to a generic MILP solver with fixed time limit. If the solver does not time out, it is guaranteed to have produced a global optimum of (2), and in Section 4.4 we find that our solver typically terminates within a practical time budget. Even if the solver times out, it will return the best feasible solution it found, plus an upper bound on the global optimal value. This bound can be used to evaluate the level of *potential* sub-optimality of the feasible solution. Note that solvers often find an optimal solution before finding the upper bound that guarantees its optimality, so timing out do not imply sub-optimality. Finally, inner-loop optimality guarantees do not translate into guarantees for the overall black-box optimization, particularly when $f(x)$ does not belong to $\mathcal{F}$. However, they do provide a useful empirical tool for understanding the impact of exact inner-loop optimization (Section 4).

## 4 Experiments

This section presents experimental results on a wide range of discrete black-box problems, with and without combinatorial constraints. We focus primarily on analyzing the effect of *global* optimization of the acquisition function, by including controlled ablations of *NN+MILP* where the inner-loop solver is replaced by an inexact evolutionary alternative. Depending on the problem, we also include independent baselines tailored to the application domain.

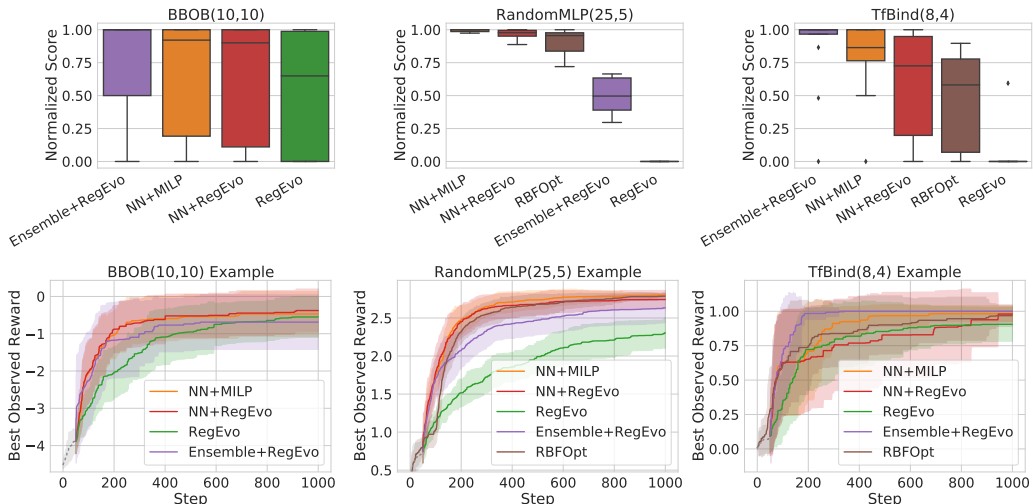

Figure 1: (Top) Distribution of algorithms' normalized scores (Section 4.1) on all unconstrained problems split by objective class. Higher is better. NN+MILP matches or outperforms NN+RegEvo on 22/30 problems. An alternate plot where scores correspond to area under the best-observed reward curve (AUC) can be found in Appendix H. (Bottom) Best observed reward as a function of iteration for an example problem in each class, averaged over 20 trials (bands indicate $\pm$1sd). Dashed grey lines in the first 50 steps indicate the initial randomly sampled dataset, common to all methods except RBFOpt, which performs its own initialization.

In all experiments, we fix the surrogate model hypothesis class $\mathcal{F}$ to networks with a single, fully-connected hidden layer of 16 neurons. Models are trained with TensorFlow (Abadi et al., 2016), using the ADAM optimizer. No hyper-parameter tuning is performed across problems. The MILP acquisition problem is solved with the Mixed-Integer Programming solver SCIP 7.0.1 (Gamrath et al., 2020) using default settings and a time limit of 500 seconds. We use standard CPU machines with ~1G RAM and $\leq$ 10 cores.

## 4.1 BENCHMARKING TASKS

Unless otherwise stated, tasks' domains consist of discrete decision vectors of length $n$, with a common alphabet $\mathcal{A}$ for all elements. We consider three families of black-box objectives:

- **RandomMLP** The output of a multi-layer perceptron operating on a one-hot encoding of the input. Notably, architectures have significantly more layers/parameters than the 16-neuron networks used as surrogates by *NN+MILP*.
- **TfBind** Binding strength of a length-8 DNA sequence to a given transcription factor (Barrera et al., 2016).
- **BBOB** Non-linear function from the continuous Black-Box Optimization Benchmarking library (Hansen et al., 2009), where each coordinate has been uniformly discretized along its range. Despite the underlying continuous structure, inputs are treated as unordered and categorical.

We use parentheses after the family name to denote dimensionality of a problem, e.g. *RandomMLP(10,5)* refers to a RandomMLP objective over a discrete domain with $n = 10$ and $|\mathcal{A}| = 5$. Appendix B lists all functions considered, and provides further details on the BBOB discretization.

Algorithms are evaluated in terms of the best reward observed after 1000 queries, averaged over 20 trials per problem. Algorithms' performance is significantly influenced by the set of $x$ that are proposed in early iterations. Therefore, to reduce variance when comparing algorithms, we initialize each of the 20 trials with a different fixed dataset of 50 random points. To facilitate comparison across problems with different reward scales, the algorithms' average final rewards are min/max normalized within each problem. That is, the best (resp. worst) on-average algorithm for a given problem is assigned a score of one (resp. zero), and intermediate values express relative distance from these extremes. No hyper-parameter tuning was performed across problems for any algorithm.

## 4.2 UNCONSTRAINED OPTIMIZATION

Before considering problems with combinatorial white-box constraints, we first tackle simple problems with no additional constraints on the discrete domain, i.e., $\Omega = \mathcal{A}^n$. This allows us to compare against general-purpose algorithms for unconstrained discrete black-box optimization. We vary the problem sizes over 30 functions, consisting of eight *RandomMLP(25,5)*, ten *BBOB(10,10)* and twelve *TfBind(8,4)* targets (Appendix B).

NN-MILP provides an analytical tool for understanding the relative impacts of the choice of surrogate model and whether the acquisition problem is solved to optimality. Doing so requires ablations that vary along two axes: the family of surrogate models and the inner-loop solver. Further configuration details are provided in Appendix C.

- **RegEvo** Local evolutionary search (Real et al., 2019) using pointwise mutations of single parent sequences and crossover recombination of two parent sequences.
- **NN + RegEvo** An ablation of *NN+MILP*, with the only difference being the use of *RegEvo* in lieu of MILP for solving the acquisition problem. Here, the inner-loop solver is allowed 10k queries of the acquisition function batched over 100 rounds, and proposes the point it has visited with the highest acquisition function value. The surrogate model is fit exactly as in NN+MILP.
- **Ensemble + RegEvo** A re-implementation of the 'MBO' baseline from Angermueller et al. (2020), using an ensemble of linear and random forest regressors as the surrogate, where hyperparameters are dynamically selected at each iteration. The acquisition function is the ensemble mean and inner-loop optimization uses *RegEvo*.
- **RBFOpt** A competitive mixed-integer black-box optimization solver that uses the 'Radial Basis Function method' as a surrogate model (Costa & Nannicini, 2018).

Figure 1 plots the distribution of algorithms' scores for all unconstrained problems and an example reward curve from each class. We omit RBFOpt from the BBOB problems since it proposes the integer midpoint (rounded down) as part of its initialization, which is close to optimal by design (see Appendix B.3). We observe that relative performance of algorithms varies significantly by objective family, with *NN+MILP* performing well across the board. In particular, we wish to highlight the empirical benefits of global optimization of the acquisition function, as illustrated by the improved performance of *NN+MILP* vs. *NN+RegEvo*. The only difference between the two is the former's stronger optimality guarantees when solving the acquisition problem. We observe that *NN+MILP* obtains a greater or equal score than its evolution-based counterpart in 22 of the 30 problems considered, and variance in its normalized scores is lower within a given objective family.

The comparison of *NN+MILP* and *Ensemble+RegEvo* solver is also instructive. Here, the primary difference is the hypothesis class $\mathcal{F}$. The strong performance of *Ensemble+RegEvo* on TfBind, and to a lesser extent BBOB, suggests that ensembles of linear and tree-based regressors are better suited to approximate those black-box objectives. However, the combination of a single neural network surrogate and exact optimization is able to achieve comparable performance.

## 4.3 CONSTRAINED OPTIMIZATION

Next, problems are augmented with combinatorial constraints on the domain. We simulate fine-balance constraints in observational studies (Zubizarreta et al., 2018; Bennett et al., 2020), where the same number of items must be selected from given sub-populations (e.g., groups sharing a common attribute). These simple, yet highly combinatorial, constraints allow for fair comparison with evolutionary algorithms that are designed to maintain feasibility with every mutation. Appendix F includes additional results on binary quadratic optimization problems with more complex constraints (e.g. graph partitioning or quadratic assignment) from the MINLPLib benchmark (Vigerske, 2021).

We use a binary alphabet $\mathcal{A} = \{0, 1\}$ to indicate whether each of $n = 100$ items is selected. These have been partitioned into given subsets $S_1, ..., S_{2k}$ for some integer $k$ and constraints enforce that the number of selected items is equal in pairs of subsets: $\sum_{i \in S_{2j-1}} z_{i1} = \sum_{i \in S_{2j}} z_{i1}$ for $j \in [k]$. We use equally-sized subsets, each of cardinality $\frac{n}{2k}$, creating 30 problems over a grid of ten objectives and three values of $k \in \{5, 10, 25\}$. Further details can be found in Appendix B. RandomMLP(100,2) functions then simulate non-linear reward structures for a given selection. A different objective class based on the Ising model is also considered in Appendix E.

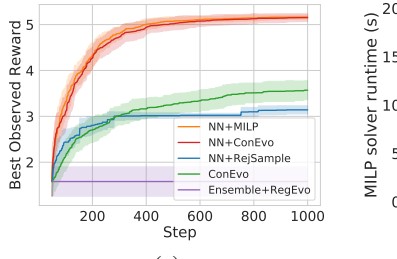 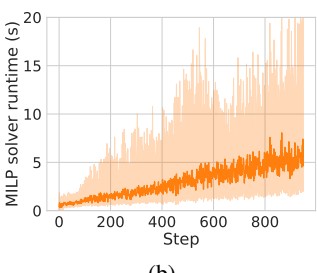 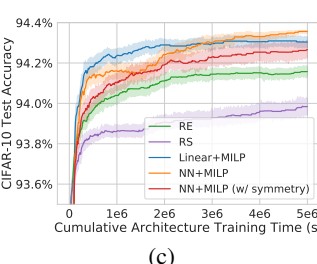

| (a) | (b) | (c) |
|---|---|---|

Figure 2: (a) Best observed reward as a function of iteration for typical constrained problem (Section 4.3), averaged over 20 trials (bands indicate $\pm$1sd). Initial randomly sampled set of 50 points is omitted. Distribution of normalized final scores and more examples can be found in Appendix H. (b) Distribution of MILP acquisition problem solve times as a function of iteration. Line and bands show the median and 5th/95th percentile range over all trials of all TfBind(8,4) problems. (c) Test accuracy of algorithms' incumbent architecture as a function of cumulative training time on NAS-Bench-101, averaged over 100 trials. Bands indicate 95% confidence interval for the mean.

The following optimization approaches provide ablations to contrast declarative vs. procedural approaches to handling the constraints. Further configuration details are given in Appendix C.

- **ConEvo** *RegEvo* with a custom mutator that procedurally maintains feasibility. Paired subsets are mutated jointly, ensuring that the number of changes in each pair is the same.
- **NN+ConEvo** An ablation of *NN+MILP* where *ConEvo* replaces MILP as the inner-loop solver. The inner-loop solver is allowed 10k queries of the acquisition function, batched over 100 rounds.
- **NN+RejSample** An ablation of *NN+MILP* where the inner-loop solver samples 10k feasible points uniformly-at-random from the domain. By using rejection sampling, the solver can leverage the same declarative definition of the constraints as in *NN+MILP*.
- **Ensemble+RegEvo** Same as Section 4.2, using the constraint definition to assign negative rewards to infeasible proposed points, which in turn affect the surrogate model.

Figure 2a plots algorithms' best observed reward as a function of iteration for a typical problem. Here, we observe that *NN+MILP* and *NN+ConEvo* perform similarly, both benefiting from the ability to model the objective with a surrogate, unlike *ConEvo*. The poor performance of *NN+RejSample* and *Ensemble+RegEvo* highlight the importance of exploiting combinatorial structure. Moreover, we obtain qualitatively similar results for an objective based on the Ising model (Appendix E).

Finally, while we did not find the difference of *NN+MILP* and *NN+ConEvo* to be statistically significant in terms of optimization performance, they differ considerably in terms of ease of implementation. In particular, *NN+MILP* required very few lines of extra code to add subset-equality constraints to the existing MILP formulation, and could have just as easily been extended to other, possibly interacting, MILP-representable constraints. Conversely, *NN+ConEvo* relied on the implementation of a custom mutator, tailored to the constraint structure at hand, and would likely require significant reworking if other constraints were added.

## 4.4 PRACTICALITY OF MILP

Figure 2b plots the distribution of MILP solve times for inner-loop optimization as function of iteration over all trials of all TfBind problems. Despite the computational complexity of the acquisition problem, MILP finds globally optimal solutions in seconds. Averaging across all unconstrained experiments (Section 4.2), the inner-loop optimization took $7.92 \pm 4.23$s for *NN+MILP* compared to $9.00 \pm 1.94$s for *NN+RegEvo* (avg. $\pm$ sd). We observed no MILP solves that exceeded the 500s time budget, and thus they were all provably optimal. As is often the case with MILP, the relationship between problem size and runtime can be unpredictable. For example, we encountered higher average solve times for lower-dimensional TfBind problems than for RandomMLP and BBOB. We also explore the impact of surrogate model network size on solve time in Appendix H. Solve times tend to increase over time, possibly due to the increasing number of no-good constraints and the nature of surrogate models that have been fit on more data.

## 5 NAS-BENCH-101 CASE STUDY

Finally, we use the NAS-Bench-101 (Ying et al., 2019) neural architecture search (NAS) benchmark to illustrate the power of MILP's declarative constraint language in formulating complex combinatorial domains. The optimization domain consists of directed acyclic graphs (DAGs) representing the *cell* in a neural architecture. Two nodes represent the input and output, and must be connected by a directed path, while the remaining nodes are each assigned to be 1x1 convolution, 3x3 convolution, or 3x3 max-pooling. Edges specify the flow of activations between nodes. The objective $f(x)$ is out-of-sample image classification accuracy. More details can be found in Appendix G.

### 5.1 MILP FORMULATIONS FOR GRAPH SEARCH

We next introduce a novel MILP formulation that precisely characterizes the set of valid NAS-Bench-101 cells. We use two sets of decision variables; the first set are binary and encode the upper-triangular adjacency matrix of a DAG with exactly $V$ nodes. The second set are a one-hot binary encoding of nodes' operations. Crucially, we introduce a new "null" operation, allowing the MILP to represent DAGs with fewer than $V$ nodes. Constraints enforce that all non-null nodes appear on a path from the input to output node, and that there exists at least one such path. A full formulation in terms of linear constraints appears in Appendix G.

A second formulation, nearly identical but enforcing so-called *symmetry-breaking* constraints, seeks to address isomorphisms in our representation. The fixed ordering of nodes above results in distinct representations for isomorphic graphs, which may be redundantly proposed despite the no-good constraints. In response, we add constraints to the MILP enforcing that null nodes must occur after any non-null nodes in the linear ordering of $x$, which removes some, but not all, isomorphisms. An alternative approach uses an isomorphism-invariant surrogate (Wen et al., 2020), but MBO based on such a model is still vulnerable to proposing isomorphic $x$.

### 5.2 RESULTS

We refer to an optimizer using the first formulation as *NN+MILP* and the second as *NN+MILP (w/ symmetry)*. Otherwise, we use the same configuration as in Section 4. *Linear+MILP* replaces the neural network surrogate with a linear model trained on $\mathcal{D}_{t-1}$ with randomization provided by bootstrapping that training data. Regularized evolution (*RE*) and random search (*RS*) baselines are from Ying et al. (2019).

Figure 2c plots the out-of-sample accuracy of the proposed architecture with the highest observed validation accuracy (the "incumbent" architecture) vs. the cumulative architecture training time. *NN+MILP*, despite its more general design, significantly outperforms *RE*. Surprisingly, we observe that both methods using no symmetry constraints, *NN+MILP* and *Linear+MILP*, perform better than *NN+MILP (w/ symmetry)*, despite optimizing over a larger search space. Finally, note that *Linear+MILP* outperforms *NN+MILP* in early iterations, but is eventually overtaken. Future work could select among MILP-compatible models at each iteration.

## 6 CONCLUSION AND FUTURE WORK

In this work we propose the *NN+MILP* framework for discrete MBO, using neural networks with ReLU activations for surrogate modeling and MILP to solve the acquisition problem. A major advantage of our method is its generality, using MILP's versatile declarative constraint language to address domains that might otherwise require specialized search algorithms for inner-loop optimization. Our experiments show that *NN+MILP* performs well on a range of discrete black-box problems with practical computational overhead using standard packages and hardware.

For future applications to continuous or mixed-integer domains, the question arises as to how to best avoid redundant proposals and encourage exploration given that no-good constraints cannot be applied as stated. More complex acquisition functions could also be considered, as long as they remain MILP-representable. For example, Expected Improvement defined over the posterior predictive distribution of an ensemble of neural networks is piecewise-linear.

## ETHICS STATEMENT

In this paper, we introduce a general algorithm for constrained discrete black-box optimization that could be applied in any number of application domains, e.g. engineering system design, neural architecture search, or drug design. As with any optimization problem, it is crucial to formulate the objective and constraints of the problem to ensure they reflect ethical design principles. Users should understand whether optimizing for a given objective might implicitly result in biased/discriminatory solutions, or have unintended consequences for some unmodeled objective. For example, in the NAS case study of Section 5 we follow common practice by maximizing out-of-sample accuracy, which might result in larger networks that are environmentally costly to train. To address this, users of our algorithm might want to incorporate environmental or ethical considerations in the (black-box) objective or constraints.

The authors do not have any conflicts of interest to disclose.

## REPRODUCIBILITY STATEMENT

We include detailed implementation sections in the Appendix to accompany all of our experiments. In particular, Appendix A describes the MILP formulation and solution techniques used for solving the *NN+MILP* acquisition problem. Appendix B contains descriptions of the *RandomMLP, TfBind* and *BBOB* benchmarks we consider in Sections 4.2 and 4.3, including how we selected problems and what hyper-parameters we used. Appendix C lists implementation details and hyper-parameter settings for *NN+MILP*, as well as all baseline algorithms from Sections 4.2 and 4.3 (*RegEvo, NN+RegEvo, Ensemble+RegEvo, RBFOpt, ConEvo, NN+ConEvo, NN+RejSample*). Finally, Appendix G contains the full MILP formulation for the NAS-Bench-101 graph search domain, as well as relevant details on the problem setting and algorithm evaluation.

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

## A  STRENGTHENING THE MILP FORMULATION FOR NEURAL NETWORKS

Here we discuss more advanced techniques for formulating the neural network surrogate model in the MILP problem. Recall the ReLU formulation constraints (4) and (5) from Section 3.3, except that we consider $M$ separately for each constraint:

$$0 \leq y \leq M_0 \alpha \tag{4'}$$

$$w^\top x + b \leq y \leq w^\top x + b + M_1(1 - \alpha), \tag{5'}$$

Here, we require a nonnegative value $M_0$ such that the right-hand side of (4') is greater or equal than a valid upper bound on $y$ when $\alpha = 1$. Similarly, $M_1$ must be a nonnegative value such that the right-hand side of (5') is greater or equal than zero when $\alpha = 0$. Therefore, we may choose $M_0$ to be any upper bound of $\max_{x \in \Omega'} w^\top x + b$ and $M_1$ to be any upper bound of $\max_{x \in \Omega'} -(w^\top x + b)$, where $\Omega'$ is the domain of the inputs of this ReLU, which depends on $\Omega$. The tighter these bounds are, the better the MILP performs.

Moreover, if we find negative $M_0$ or $M_1$, then we may (in fact, must) replace the formulation by $y = 0$ or $y = w^\top x + b$ respectively, since in these cases the ReLU is always inactive or active for any $x \in \Omega'$. This replacement must be done because the formulation assumes nonnegative $M_0$ and $M_1$ for feasibility.

The simplest way to compute $M_0$ and $M_1$ is to start from bounds in $\Omega$ and propagate them via interval arithmetic. For example, if $x \in [L, U]$, then $M_0$ can be set to $\sum_{i:w_i>0} w_i U_i + \sum_{i:w_i<0} w_i L_i + b$ and $M_1$ to $-(\sum_{i:w_i>0} w_i L_i + \sum_{i:w_i<0} w_i U_i + b)$. However, despite being fast, the drawback of this simple approach is that it does not take into account constraints on $\Omega$ or one-hot and no-good constraints.

In our experiments, we compute $M_0$ and $M_1$ by solving the linear programming (LP) relaxations of $\max_{x \in \Omega'} w^\top x + b$ and $\max_{x \in \Omega'} -(w^\top x + b)$ respectively (i.e. without integrality constraints). We remark that for neurons in the same layer these LPs have the same constraints but different objectives, and thus we may take advantage of the warm starting functionality in LP solvers. While this requires solving two LPs per neuron, taking into account the constraints from $\Omega$ into the bounds often enable the overall MILP to be solved much faster.

The formulation can also be strengthened with cutting plane techniques (Anderson et al., 2020), but they are not particularly beneficial for the small network sizes considered in this paper (at most a single layer with 16 ReLUs) and thus we do not add them. Future work could explore warm-starting the MILP solver using results from earlier MBO iterations.

## B  BENCHMARKING TASKS

This section details the black-box objective functions considered in both unconstrained (Section 4.2) and constrained (Section 4.3) experiments. Recall that all objective functions are defined over fixed-length discrete vectors of length $n$, with each element drawn from an alphabet $\mathcal{A}$ of fixed size.

### B.1  TFBIND

The objective function is given by the binding affinity of a length-8 DNA sequence to a particular transcription factor, characterized experimentally in the dataset described by Barrera et al. (2016). The problem size is thus fixed by the application at hand, with $n = 8$ and $|\mathcal{A}| = 4$ (each input element corresponding to a given DNA nucleotide). We min/max-normalize the binding affinity values for each factor to the zero-one interval. We create 12 unconstrained problems (Section 4.2) using the following datasets: CRX_R90W_R1, CRX_REF_R1, FOXC1_REF_R1, GFI1B_REF_R1, HOXD13_Q325R_R1, HOXD13_REF_R1, NR1H4_C144R_R1, NR1H4_REF_R1, PAX4_REF_R1, PAX4_REF_R2, POU6F2_REF_R1, SIX6_REF_R1. Here, the 3 fields separated by underscores represent the transcription factor id, any mutations that have been made to the transcription factor, and the id of the experimental replicate used when collecting data.

## B.2 RANDOMMLP

The objective function is given by the output of a multi-layer perceptron (MLP) with randomly-sampled weights. Different functions are generated by varying the architecture type (described below) and random seed. All architectures employ a one-hot encoding of the inputs as the first layer. Weights are sampled using the default behavior of tf.keras.layers.Dense (`glorot_uniform`).

We consider two architecture types, both utilizing more layers/parameters than the 16-neuron networks used by *NN+MILP* (Section 4). The *RandomFCC* architecture uses two fully-connected layers with 128 hidden units each, while the *RandomCNN* architecture uses two convolutional layers each with 64 hidden units each, a kernel width of 13 and stride size of 1. We use a linear activation function for the output and ReLU activations for all intermediate layers.

Unconstrained *RandomMLP* problems (Section 4.2) all have size $n = 25$ and $|\mathcal{A}| = 5$. Eight objective functions are created by varying the architecture type (FCC or CNN) and random seed (0, 13, 42, 77). Constrained *RandomMLP* problems (Section 4.3) all have size $n = 100$ and $|\mathcal{A}| = 2$. Thirty problems are created by varying the architecture type (FCC or CNN), random seed (0, 13, 42, 77, 100) and the number of paired subsets (5, 10, 15) in the subset-equality constraints (defined in Section 4.3).

## B.3 BBOB

The objective is given by a function from the continuous Black-Box Optimization Benchmarking library (Hansen et al., 2009). All BBOB functions are defined for a variable number of dimensions $n$ and the search domain is given as $[-5, 5]^n$, with the global optimum centered at zero. We normalize each function's output range by evaluating it at 30 fixed points and dividing outputs by the median absolute deviation in those points' values.

We discretize functions for our setting (Section 3.1) by defining a grid over the continuous search domain, adjusted so that the optimal solution exactly corresponds to a point in the grid. Concretely, we use a fixed alphabet $\mathcal{A} = \{1, \dots, m\}$ for all coordinates, denoting the *index* of one of $m$ allowed values for that coordinate. Allowed values for each coordinate are $m$ equally-spaced points in the range $[-5, 5]$, except for a point lying closest to zero which is overwritten to exactly equal that value. In this way, the optimum is guaranteed to lie on the discretized grid. Note that, despite the underlying continuous structure, all algorithms treat each dimension as an unordered, categorical variable.

For unconstrained *BBOB* problems (Section 4.2), we select a diverse set of objectives by taking two functions from each of the five categories defined by the BBOB library:

1. Separable functions: Sphere (SPHERE) and Ellipsoidal (ELLIPSOID_SEPARABLE).
2. Functions with low or moderate conditioning: Attractive Sector (ATTRACTIVE_SECTOR) and Step Ellipsoidal (STEP_ELLIPSOID).
3. Functions with high conditioning and unimodal: Discus (DISCUS) and Bent Cigar (BENT_CIGAR).
4. Multi-modal functions with adequate global structure: Weierstrass (WEIERSTRASS) and Schaffers F7 (SCHAFFERS_F7).
5. Multi-model functions with weak global structure: Schwefel (SCHWEFEL) and Gallagher's Gaussian 21-hi Peaks (GALLAGHER_21ME).

We set the dimension for all of these to $n = 10$ and discretize as described above, using an alphabet of size $|\mathcal{A}| = 10$ for all coordinates. We purposefully use a relatively large alphabet to ensure that the discretization does not obscure any inherent variance across a given coordinate.

## C  BASELINE OPTIMIZATION ALGORITHMS

In this section we describe implementation and configuration details for all baseline optimization algorithms described in Section 4.

## C.1   NN+MILP

For our experiments, we implement our main algorithm (Section 3) as follows: we use a fixed surrogate model hypothesis class $\mathcal{F}$ of networks with a single, fully-connected hidden layer of 16 neurons. Models are trained with TensorFlow (Abadi et al., 2016), using the ADAM optimizer for $25K$ epochs with a batch size of 64 and no explicit regularization. We use a constant learning rate of $\alpha = 0.01$ and default decay parameters $(\beta_1, \beta_2) = (0.9, 0.999)$. No hyper-parameter tuning is performed across problems. Model training is randomized due to the random example ordering of SGD training and random parameter initialization. The MILP acquisition problem is solved with the Mixed-Integer Programming solver SCIP 7.0.1 (Gamrath et al., 2020) using default settings and a time limit of 500 seconds. In order to increase the diversity of trained models, we train each model from scratch at each iteration of optimization instead of fine-tuning a model from an earlier iteration.

## C.2   REGEVO

We re-implement the local evolutionary search algorithm of Real et al. (2019), and extend the set of mutation operators from just pointwise mutators to also include a crossover operation that recombines two parent sequences. The algorithm proposes $x_{t+1}$ by selecting two parent sequences from the existing population, recombining them and mutating them. Parents are chosen by tournament selection, taking the two best samples from a randomly-selected subset of size $T$ of previously sampled points. The pool from which parents can be selected is limited to the $D$ most recently-proposed points (referred to as the "alive population"), to avoid high-reward points from early rounds dominating the process. The selected parent sequences are recombined by copying them left-to-right, starting a pointer at one parent at switching reading to the other parent with a fixed cross-over probability $p_c$ after each copy. The resulting sequence is finally mutated by changing each position to a different token from $\mathcal{A}$ with a fixed probability $p_m$.

In the unconstrained experiments (Section 4.2), we use *RegEvo* as the outer-loop optimization algorithm and set the tournament size to $T = 10$, the alive population size to $D = 100$, and the crossover/mutation probabilities to $(p_c, p_m) = (0.1, 0.1)$.

## C.3   NN+REGEVO

This algorithm is an ablation of *NN+MILP*, with the only difference being the use of *RegEvo* in lieu of MILP to solve the acquisition problem at every iteration. A surrogate neural network $\hat{f}_t \in \mathcal{F}$ is trained as in *NN+MILP*, and the acquisition function is $a(x) = \hat{f}_t(x)$. The problem of selecting $x_{t+1}$ is posed as a *batched* optimization problem and solved by *RegEvo*.

More concretely, at iteration $t$, the acquisition function is evaluated for all points in the existing population $\mathcal{D}_t$ to generate the initial inner-loop population $\hat{\mathcal{D}}_t := \{x_i, a(x_i)\}$. This population is iteratively extended by generating candidate proposals with *RegEvo* in batches of size $b$, and with rewards now corresponding to the value of the acquisition function rather than the original black-box function. That is, *RegEvo* generates $b$ points by recombination/mutation of parents from $\hat{\mathcal{D}}_t$, which are evaluated on the acquisition function and added to the inner-loop population. The process repeats until a total of $B$ candidates have been generated, at which point the one with the highest acquisition function value (excluding any points already proposed) is proposed as $x_{t+1}$.

In the unconstrained experiments (Section 4.2), we use *NN+RegEvo* and set surrogate model hyper-parameters exactly as in *NN+MILP* (Section C.1). For the inner-loop optimizer, we set the total number of acquisition function evaluations to $B = 10,000$ and batch size to $b = 100$. The *RegEvo* optimizer's hyper-parameters, defined in Section C.2, are set to $T = 20$, $D = 1,000$ and $(p_c, p_m) = (0.2, 0.01)$.

## C.4   ENSEMBLE+REGEVO

We recreate the *MBO* baseline of Angermueller et al. (2020). Here, surrogate modeling proceeds by optimizing the hyper-parameters of a diverse set of regressor models through randomized search. Regressors are trained using the `scikit-learn` library (Pedregosa et al., 2011), drawing from the following model classes (randomized search parameters are listed in parentheses):

- LassoRegressor (alpha)
- RidgeRegressor (alpha)
- RandomForestRegressor (max_depth, max_features, n_estimators)
- LGBMRegressor (learning_rate, n_estimators)

Each model is evaluated by an explained variance score using five-fold cross validation on the training set. All models with a score $\geq 0.4$ are used as an ensemble for the surrogate model, with their *average* prediction serving as the acquisition function. The acquisition problem is solved by batched *RegEvo* with a total of $B = 12,500$ acquisition function evaluations and a batch size of $b = 25$. The optimizer's hyper-parameters, defined in Section C.2, are set to $T = 20$, $D = 1,000$ and $(p_c, p_m) = (0.2, 0.01)$.

We use *Ensemble+RegEvo* in both the unconstrained (Section 4.2) and constrained (Section 4.3) experiments. In the latter case, we use the algorithm as a baseline that makes use of the declarative definition of constraints; during training of the ensemble, infeasible points are assigned a highly negative reward (worse than any observed). In this way, the surrogate model might be expected to implicitly model infeasibility with low predictions which should be avoided by the inner-loop optimizer.

## C.5 RBFOPT

*RBFOpt* (Costa & Nannicini, 2018) is a black-box optimization solver for mixed-integer unconstrained problems (i.e. with only bound constraints) that performs competitively with respect to other solvers of its type. It uses a Radial Basis Function as a surrogate model and includes a number of practical enhancements. It relies on a mixed-integer nonlinear programming (MINLP) solver, BONMIN (Bonami et al., 2008), to optimize the inner loop problems. The MINLP solver could in theory incorporate constraints in a similar fashion as in our work, although this is not offered by the open-source implementation (aside from manually penalizing the objective function) and we expect it to not scale as well as a MILP solver in practice since MINLP is a significantly more difficult problem class than MILP.

We use *RBFOpt* for our unconstrained experiments (Section 4.2), using the open-source implementation available at `https://github.com/coin-or/rbfopt`. We leave all settings at their defaults, including building the initial set of points. We note in particular that the API for this implementation uses an integer encoding for categorical variables (constraints are not supported, which precludes a one-hot representation). As we note in the main text, we omit the RBFOpt results for BBOB because RBFOpt proposes the midpoint of this integer representation (rounded down) as part of its initialization, which is close to the optimal solution.

## C.6 CONEVO

In Section 4.3 we introduce *ConEvo*, a local evolutionary search algorithm that exploits the known combinatorial structure of the subset-equality constraints considered therein. The method selects just a single parent sequence (using the same tournament procedure as *RegEvo*) and mutates it in a way that guarantees feasibility of the child sequence. We do not implement recombination of multiple parent sequences since they are not likely to maintain feasibility. We described the application-specific mutator below.

Recall that the domain encodes the selection or not of each of $n$ items using a binary alphabet $\mathcal{A} = \{0, 1\}$. The items' indices are partitioned into disjoint, equally-sized subsets $S_1, \ldots, S_{2k}$ for some $k$ and the constraints enforce that the number of selected items should be the same in pairs of subsets; that is:

$$\sum_{i \in S_{2j-1}} \mathbb{I}\{x_i = 1\} = \sum_{i \in S_{2j}} \mathbb{I}\{x_i = 1\} \quad \forall j \in [k]$$

where we have used indicator notation and the original decision variables $x$ rather the one-hot encoding.

The mutator begins with a single parent sequence $x$, assumed feasible, and is given access to the item subsets $S_1, \ldots, S_{2k}$. Each pair of subsets $(S_{2j-1}, S_{2j})$ is mutated concurrently to create the child sequence $y$, ensuring that mutations to one subset are counter-balanced by mutations to the second. Concretely, one of the two subsets is chosen randomly to be the "independent" mutatee with equal probability. We denote the selected subset $\mathcal{I}^+$, and the other subset in the pair by $\mathcal{I}^-$. Each position $i \in \mathcal{I}^+$ of the child sequence is flipped from its parent value with some fixed probability $p_m$. We compute $c$, the *net* number of 0-to-1 conversions in positions $\mathcal{I}^+$. If $c$ is positive (i.e. there were more 0-to-1 conversions than 1-to-0 conversions) then exactly $c$ indices are chosen randomly from $\{i \in \mathcal{I}^- : x_i = 0\}$, and also flipped in the child. If $c$ is negative, then $-c$ indices are selected randomly from $\{i \in \mathcal{I}^- : x_i = 1\}$ and flipped in the child. If $c$ is zero, the positions in $\mathcal{I}^-$ are left unchanged in the child. As a result, the subsets $S_{2j-1}$ and $S_{2j}$ retain exactly the same number of selected items in the mutated sequence.

In the constrained experiments (Section 4.3), we use *ConEvo* as the outer-loop optimization algorithm setting the tournament size for selecting parent sequences to $T = 20$ and the mutation probability to $p_m = 0.05$.

### C.7 NN+CONEVO

This algorithm is an ablation of *NN+MILP* used in Section 4.3, with the only difference being the use of *ConEvo* in lieu of MILP to solve the *constrained* acquisition problem at every iteration. A surrogate neural network $\hat{f}_t \in \mathcal{F}$ is trained as in *NN+MILP*, and the acquisition function is $a(x) = \hat{f}_t(x)$. Selecting $x_{t+1}$ is posed as a *batched* optimization problem and solved by *ConEvo*. The batched optimization procedure is exactly as described for *NN+RegEvo* (Section C.3).

In the constrained experiments (Section 4.3), we use *NN+ConEvo* and set surrogate model hyper-parameters exactly as in *NN+MILP* (Section C.1). For the inner-loop optimizer, we set the total number of acquisition function evaluations to $B = 10,000$ and batch size to $b = 100$. The *ConEvo* optimizer's hyper-parameters, defined in Section C.6, are set to $T = 20$ and $p_m = 0.05$.

### C.8 NN+REJSAMPLE

This algorithm is an ablation of *NN+MILP* used in Section 4.3, with the only difference being the use of random search in lieu of MILP to solve the *constrained* acquisition problem at every iteration. A surrogate neural network $\hat{f}_t \in \mathcal{F}$ is trained as in *NN+MILP*, and the acquisition function is $a(x) = \hat{f}_t(x)$.

To select $x_{t+1}$, we first generate $B$ points uniformly-at-random from the constrained domain via rejection sampling. That is, candidates are drawn uniformly at random from $\mathcal{A}^n$ and rejected if they do no satisfy the subset equality constraints (Section 4.3). The process continues until $B$ feasible points have been generated, and the one with the highest observed acquisition function value (excluding any points already proposed) is proposed as $x_{t+1}$.

In the constrained experiments (Section 4.3) we use *NN+RejSample* and set surrogate model hyper-parameters exactly as in *NN+MILP* (Section 4). We set the number of points to generate at each optimization step to $B = 10,000$. Empirically, we observe a rejection rate of $\approx 91\%$ for the subset-equality constraints in our experiments.

## D BINARY VS INTEGER VARIABLES

In this work, we focus on problems with binary domain formulations (e.g. one-hot encoding of categorical domains), and even problems with integer variables such as the discretized BBOB are binarized. Part of the reason is to allow no-good constraints as described in Section 3.3, but in addition we have experimentally observed that the method performs better when using a binary encoding instead of an integer one.

When running this algorithm for unconstrained (bounded) integer or continuous problems, we have informally observed that our method frequently proposes solutions where several variable values are at either their lower bound or upper bound. As a result, our method would underexplore solutions away from the boundary. A possible explanation for this is that feedforward ReLU networks tend

to extrapolate linearly, and thus their optima may often lie on the boundary (Xu et al., 2021). In contrast, every feasible point of a binary problem lies on a corner of the 0-1 hypercube. A similar observation has been made in the context of IDONE (Bliek et al., 2021), which also uses a ReLU-based surrogate model: encoding the Rosenbrock problem using binary variables improves the performance of the IDONE algorithm, although the opposite happens for a Bayesian optimization algorithm (Karlsson et al., 2020).

We provide computational evidence of this behavior in Figure 3 for TfBind and BBOB instances. The binary variables are encoded as one-hot variables, whereas the integer variables follow an arbitrary ordering for TfBind and the problem ordering for BBOB.

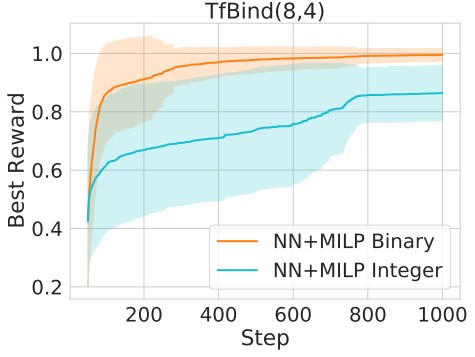 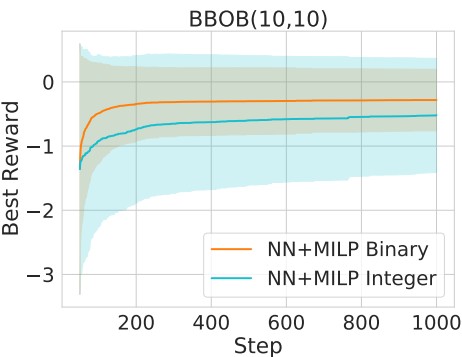

Figure 3: Best observed reward as a function of iteration for all TfBind (left) and BBOB (right) instances, comparing the use of binary and integer variables. For TfBind, the categorical variables are transformed to integer with an arbitrary ordering, and for BBOB, we use the given ordering of the problem. Note that the error region is large here since we aggregate all of the instances of each class.

# E  CONSTRAINED OPTIMIZATION WITH AN ISING PROBLEM

To provide further evidence that the results from Section 4.3 can generalize beyond the RandomMLP function, Figure 4 shows the result for the same set of constrained experiments with the only difference that we switch the RandomMLP objective by another random function based on the Ising model. Here, the Ising model represents a fully-connected graph with nodes that can take binary values. Each edge is defined by a $2 \times 2$ table of scores for each possible configuration of the nodes that are connected by the edge. Each value in the tables is drawn from a normal distribution and the overall function is the sum of the scores over all edges. We aggregate over 10 random instances and 20 replications with different random seeds. We omit Ensemble+RegEvo with penalty as described in Section 4.3 since it does not perform well with these constraints. We observe that the results are qualitatively similar to those of Section 4.3: NN+MILP continues to perform significantly better than ConEvo, while NN+MILP and NN+ConEvo have similar performance.

We also experiment with the same class of constrained problems but defined over larger domains ($n = 200$ and $400$ binary decision variables). Given the larger scale, we set a single value for the number of paired subsets in the constraints, namely $k = 20$ when $n = 200$ or $k = 40$ when $n = 400$ (in other experiments, different values of $k$ resulted in qualitatively similar results), and we do not run NN+RejSample. We aggregate over 10 random instances and 10 replications for each problem size. In Figure 5, we observe that NN+MILP significantly outperforms NN+ConEvo for the $n = 400$ instances, suggesting that it is beneficial to solve the acquisition problem to optimality at larger scales.

Finally, in Figure 6 we examine the impact the surrogate model's capacity in this larger-scale setting. We include ablations of both NN+MILP and NN+ConEvo where the surrogate model has 32 neurons in the hidden layer, instead of the 16 used before. Despite optimizing over a larger domain, neither NN+MILP nor NN+ConEvo show substantial improvements in performance when using a larger

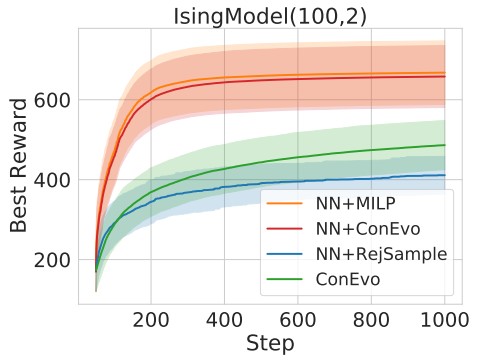 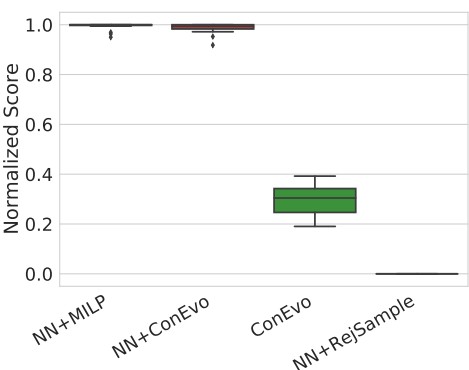

Figure 4: (Left) Best observed reward as a function of iteration for random instances based on the Ising model with the subset equality constraints described in Section 4.3. (Right) Distribution of the algorithms' normalized scores on the same constrained Ising model problems. Higher is better.

surrogate network. This suggests that, in this case at least, the relatively small number of training points is a more significant bottleneck for approximation than the capacity of the surrogate.

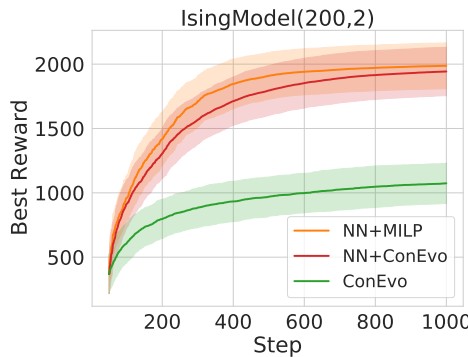 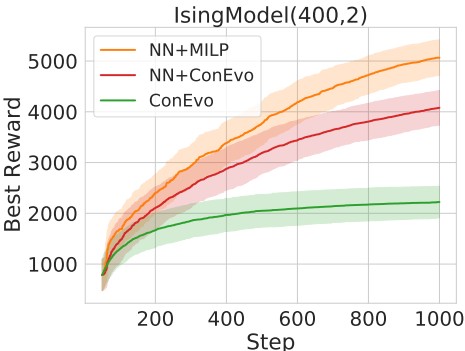

Figure 5: Best observed reward as a function of iteration for random instances based on the Ising model with 200 (left) and 400 (right) binary variables and the subset equality constraints described in Section 4.3, except that the number of pairs of subsets is $k = 20$ (left) and $k = 40$ (right).

## F CONSTRAINED BINARY QUADRATIC PROBLEMS FROM MINLPLIB

We study the performance of our method for linearly-constrained binary quadratic problems from the MINLPLib benchmark library (Vigerske, 2021). In practice, one would use a specialized mixed-integer quadratic programming solver to tackle these problems, but they serve well as a benchmark for our black-box optimization method since they are still harder to solve than MILP and include practically-motivated constraints, along with offering good feasible solutions to compare with. Many of these instances are of a slightly larger scale than typical black-box optimization problems (i.e. a few hundreds of variables), which helps us evaluate the method at larger scale and observe its scalability limitations.

We select the instances of type "BQP" from MINLPLib with at least one constraint. We discard the 11 instances prefixed by `celar6-sub0`, `color_lab`, and `max_csp`, which are large instances for which our method was unable to find a solution with primal gap at most 10% (see below). This

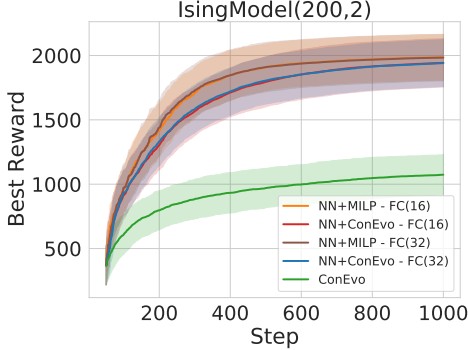
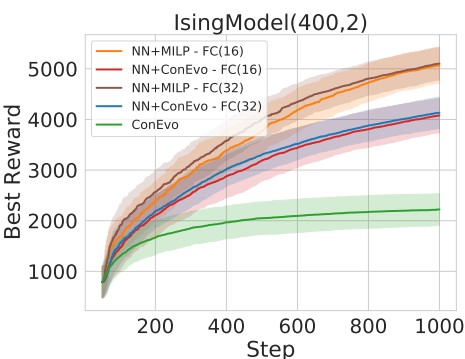

Figure 6: Best observed reward as a function of iteration for random instances based on the Ising model with 200 (left) and 400 (right) binary variables and the subset equality constraints described in Section 4.3, except that the number of pairs of subsets is restricted to $k = 10$. FC(16) and FC(32) represent the runs where the surrogate neural network has a single layer of 16 and 32 ReLUs respectively.

leaves us with the 50 instances listed in Table 1. For consistency with the remainder of the paper, we turn the problems into maximization problems by negating the objective function.

We run NN+MILP with the same settings as previous experiments (see Appendix C.1). One difference here is that the feasible set may be too small to sample from using rejection sampling. Therefore, we build our initial set of 50 points by randomly choosing an objective direction and solving an MILP under the constraints of the problem, which is practically feasible since the scale of these problems is small in the context of MILP. In our experiments, we compare with the best known primal feasible solution from the MINLPLib benchmark itself (as of October 1, 2021).

Out of the 20 runs for each instance, we show in Table 1 the number of runs that found a solution in 1000 steps with value equal to the best known value from MINLPLib, or with a primal gap of at most 1% or 10% with respect to that value. We use the definition of primal gap from Berthold (2013): if $v^*$ is the best known value and $v$ is the objective value given by our method, then the primal gap is $\frac{|v-v^*|}{\max(|v|,|v^*|)}$, or 0 if $|v| = |v^*| = 0$, or 1 if $v$ and $v^*$ have different signs.

Interestingly, in 20 out of the 50 instances, we match the best known objective value in at least one of the runs. This includes a number of instances that are considered to be large for black-box optimization, such as the general quadratic assignment problem `pb302095` which has 600 variables.

On the other hand, we also observe that the method has difficulties in finding a good solution for larger instances. This is more clearly illustrated by Figure 7, in which we observe how the method scales with the graph partitioning problems denoted by `graphpart_clique`. For the smaller instance (with 60 variables), our method finds an optimal solution in relatively few steps, but it has difficulties in reaching the best known solution for the larger instance (with 180 variables) with the same constraint structure. That said, this does not mean that this method cannot be extended to scale further (e.g. we have not attempted to tune the parameters or change the architecture of the surrogate model for larger instances).

# G  NAS-BENCH-101 CASE STUDY

## G.1  BACKGROUND

In Section 5 we consider the NAS-Bench-101 (Ying et al., 2019) neural architecture search (NAS) benchmark as a case study, to illustrate the power of MILP's declarative constraint language in modeling complex combinatorial domains. The optimization domain consists of directed acyclic

Table 1: Results for a subset of binary quadratic problems from the MINLPLib benchmark, indicating the number of runs out of 20 for which the primal gap with respect to the best known primal feasible solution is at most 0%, 1%, and 10%.

| Instance name | # variables | Number of runs with solution at | | |
|---|---|---|---|---|
| | | 0% gap | ≤ 1% gap | ≤ 10% gap |
| cardqp_inlp | 50 | 5 | 14 | 20 |
| cardqp_iqp | 50 | 5 | 14 | 20 |
| crossdock_15x7 | 210 | 0 | 0 | 20 |
| crossdock_15x8 | 240 | 0 | 0 | 20 |
| graphpart_2g-0044-1601 | 48 | 17 | 17 | 20 |
| graphpart_2g-0055-0062 | 75 | 0 | 2 | 19 |
| graphpart_2g-0066-0066 | 108 | 0 | 0 | 17 |
| graphpart_2g-0077-0077 | 147 | 0 | 0 | 7 |
| graphpart_2g-0088-0088 | 192 | 0 | 0 | 7 |
| graphpart_2g-0099-9211 | 243 | 0 | 0 | 2 |
| graphpart_2g-1010-0824 | 300 | 0 | 0 | 0 |
| graphpart_2pm-0044-0044 | 48 | 20 | 20 | 20 |
| graphpart_2pm-0055-0055 | 75 | 13 | 13 | 20 |
| graphpart_2pm-0066-0066 | 108 | 6 | 6 | 15 |
| graphpart_2pm-0077-0777 | 147 | 0 | 0 | 7 |
| graphpart_2pm-0088-0888 | 192 | 0 | 0 | 5 |
| graphpart_2pm-0099-0999 | 243 | 0 | 0 | 3 |
| graphpart_3g-0234-0234 | 72 | 0 | 4 | 18 |
| graphpart_3g-0244-0244 | 96 | 0 | 1 | 17 |
| graphpart_3g-0333-0333 | 81 | 4 | 4 | 19 |
| graphpart_3g-0334-0334 | 108 | 0 | 0 | 16 |
| graphpart_3g-0344-0344 | 144 | 0 | 1 | 8 |
| graphpart_3g-0444-0444 | 192 | 0 | 0 | 8 |
| graphpart_3pm-0234-0234 | 72 | 7 | 7 | 19 |
| graphpart_3pm-0244-0244 | 96 | 0 | 0 | 17 |
| graphpart_3pm-0333-0333 | 81 | 1 | 1 | 15 |
| graphpart_3pm-0334-0334 | 108 | 0 | 0 | 11 |
| graphpart_3pm-0344-0344 | 144 | 0 | 0 | 5 |
| graphpart_3pm-0444-0444 | 192 | 0 | 0 | 6 |
| graphpart_clique-20 | 60 | 20 | 20 | 20 |
| graphpart_clique-30 | 90 | 20 | 20 | 20 |
| graphpart_clique-40 | 120 | 13 | 13 | 18 |
| graphpart_clique-50 | 150 | 0 | 0 | 0 |
| graphpart_clique-60 | 180 | 0 | 0 | 0 |
| graphpart_clique-70 | 210 | 0 | 0 | 0 |
| pb302035 | 600 | 0 | 0 | 0 |
| pb302055 | 600 | 0 | 0 | 20 |
| pb302075 | 600 | 5 | 5 | 20 |
| pb302095 | 600 | 13 | 20 | 20 |
| pb351535 | 525 | 0 | 0 | 15 |
| pb351555 | 525 | 1 | 3 | 20 |
| pb351575 | 525 | 0 | 10 | 20 |
| pb351595 | 525 | 6 | 19 | 20 |
| qap | 225 | 0 | 0 | 7 |
| qspp_0_10_0_1_10_1 | 180 | 5 | 5 | 20 |
| qspp_0_11_0_1_10_1 | 220 | 9 | 20 | 20 |
| qspp_0_12_0_1_10_1 | 264 | 13 | 13 | 20 |
| qspp_0_13_0_1_10_1 | 312 | 0 | 19 | 20 |
| qspp_0_14_0_1_10_1 | 364 | 0 | 16 | 20 |
| qspp_0_15_0_1_10_1 | 420 | 12 | 13 | 20 |

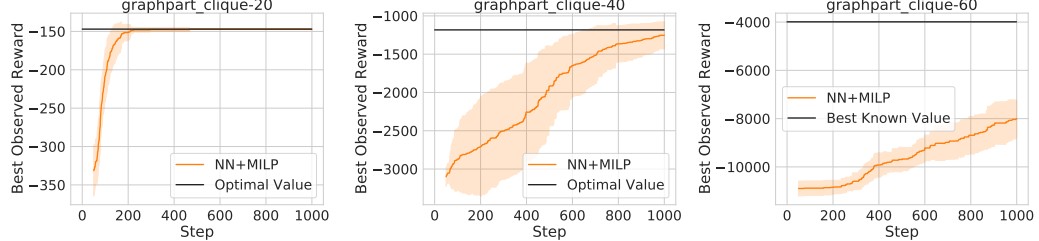

Figure 7: Best observed reward as function of iteration for three graph partitioning instances from MINLPLib (negated for maximization), with 60, 120, and 180 binary variables respectively. Black lines show the best known feasible solution to the problem (as of October 1, 2021). Colored lines show the average over 20 trials, while bands indicate $\pm 1$ sd. Note that bands that exceed the black line are an artifact of the symmetric nature of standard deviation, and do not necessarily mean a trial found an improved solution.

graphs (DAGs) with a maximum of $V = 7$ nodes and $M = 9$ edges, representing the *cell* in a neural architecture. The overall model is obtained by stacking multiple copies of the cell. Two nodes represent the input and output, and must be connected by a directed path, while the remaining nodes are each assigned to be 1x1 convolution, 3x3 convolution, or 3x3 max-pooling. Edges specify the flow of activations between nodes. The goal is to find the cell architecture that maximizes out-of-sample accuracy on a given image classification task.

NAS differs from the problem setting in Section 3.1 in three key ways. First, algorithms do not have access to the true objective (out-of-sample accuracy), but instead a correlated proxy (validation accuracy). Second, $f(x)$ is noisy due to the stochasticity of classifier training, and thus algorithms may benefit from repeated queries of the same point. Finally, algorithms may benefit by leveraging the validation accuracy at early epochs as a proxy to halt unpromising evaluations.

Despite these differences, we apply *NN+MILP* exactly as described in Section 3/Appendix C.1 (including no-good constraints which prevent repeated queries). We reimplement the Regularized Evolution (*RE*) and random search (*RS*) baselines from the original NAS-Bench-101 paper (Ying et al., 2019), using the same hyper-parameters settings specified therein.

The NAS-Bench-101 dataset contains pre-computed validation *and* test accuracies for three independently trained replications of each architecture, as well as the training time of each. To simulate NAS, algorithms' observed reward after proposing an architecture is the validation accuracy of a randomly sampled replication from said architecture. This defines the notion of an "incumbent" proposal, namely the proposed architecture with the highest (observed) validation accuracy, which may not in fact be the best (unobserved) test accuracy. Instead of allowing algorithms a fixed budget of evaluations, we use a fixed budget of $T = 5 \times 10^6$ seconds, and allow algorithms to query the objective until cumulative training time exceeds the budget. For evaluation purposes (e.g. Figure 2c) we plot the out-of-sample accuracy of the incumbent architecture as a function of cumulative architecture training time.

## G.2 DOMAIN FORMULATION

To formulate the NAS-Bench-101 domain, we first define a representation of cell architectures as fixed-length binary vectors. We split the representation into two components; one set of variables encodes the presence or absence of each graph edge, while the second is a one-hot encoding of nodes' assigned operations. As all valid cell graphs are directed and acyclic, we limit the edge variables to the strict upper triangle of the adjacency matrix, which implicitly enforces a topological ordering of the nodes in any feasible solution and ensures acyclicity. The first- and last-indexed nodes are always assigned the `input` and `output` operations respectively, while intermediates nodes can be assigned any operation from the set $\mathcal{S} = \{\texttt{conv1x1}, \texttt{conv3x3}, \texttt{maxpool3x3}\}$.

To ensure a fixed-length set of decision variables while allowing for graphs with a variable number of nodes, we introduce a new `null` operation. Nodes assigned the null operation are not considered part of the computational graph of the cell. The algorithm then searches over the space of binary

representations, constrained to yield feasible cell architectures. Denoting by $V$ and $M$ the maximum number of allowable nodes and edges respectively, the decision variables (all binary) are:

- $m_{i,j}$ for $1 \leq i < j \leq V$, 1 if there is an edge from node $i$ to node $j$, 0 otherwise.
- $w_{i,k}$ for $1 \leq i \leq V, 1 \leq k \leq |\mathcal{S}|$, 1 if node $i$ is assigned the $k$'th operation in $\mathcal{S}$, 0 otherwise.
- $z_i$ for $1 < i < V$, 1 if node $i$ is assigned the null operation, 0 otherwise.

The feasible set of cell architectures can then be given in terms of linear constraints as follows:

$$w_{1,k} = w_{V,k} = z_1 = z_V = 0 \qquad \text{for } 1 \leq k \leq |\mathcal{S}| \tag{1}$$

$$z_i + \sum_{k=1}^{S} w_{i,k} = 1 \qquad \text{for } 1 < i < V \tag{2}$$

$$\sum_{i=1}^{V} \sum_{j=i+1}^{V} m_{i,j} \leq M \tag{3}$$

$$m_{i,j} \leq 1 - z_j \qquad \text{for } 1 \leq i < j \leq V \tag{4}$$

$$m_{i,j} \leq 1 - z_i \qquad \text{for } 1 \leq i < j \leq V \tag{5}$$

$$\sum_{i=1}^{j-1} m_{i,j} \geq 1 - z_j \qquad \text{for } 1 \leq j \leq V \tag{6}$$

$$\sum_{j=i+1}^{V} m_{i,j} \geq 1 - z_i \qquad \text{for } 1 \leq i \leq V \tag{7}$$

$$z_i \leq z_{i+1} \qquad \text{for } 1 < i < V - 1 \tag{8}$$

Constraints 1 ensure that the input and output nodes are not assigned any operation from $\mathcal{S}$ or null, while 2 enforces the one-hot encoding of operations for intermediate nodes (including the possibility of a null operation). Constraint 3 imposes a limit on the number of edges in the graph, per the NAS-Bench-101 specifications. Constraints 4 & 5 assert that null nodes have no incoming or outgoing edges respectively, effectively disconnecting them from the remaining graph. Conversely, 6 & 7 assert that non-null nodes have at least one ingoing and one outgoing edge. Crucially, due to the implicit topological sorting of nodes by the upper-triangular adjacency matrix, these also ensure that there is always a path from the input to the output node using only non-null nodes. Intuitively, all non-null nodes (including the input) have at least one outgoing edge – which necessarily leads to a higher-indexed non-null node – and all non-null nodes (including the output) have at least one-incoming edge – which necessarily comes from a lower-indexed non-null node. The flow exiting the input node, must eventually enter the output node.

Finally, we focus on Constraints 8, which we refer to as *symmetry-breaking* constraints. These assert that a node can only be assigned the null operation if its topological successor has also been assigned it. While not necessary for feasibility, this constraint serves to eliminate symmetry by ensuring that all null nodes are topologically sorted after any non-null nodes. In essence, it introduces a "canonical" labeling of null vs. non-null nodes, whose isomorphic representations are excluded from the feasible region.

In our experiments, we found that including symmetry-breaking constraints actually resulted in worse overall performance for the outer optimization problem (Figure 2c). We hypothesize that this is due to a reduction in the exploration behaviour of *NN+MILP*, as the surrogate's predictive distribution was more uncertain in the larger search space and the inner-loop optimizer thus more likely to propose points in unexplored areas. One possible future line of work could be to augment $\mathcal{D}_t$ with isomorphic representations before training, e.g. by random reordering of nodes in the representations of sampled points.

# H  ADDITIONAL PLOTS

We present in this section additional figures related to the experiments in the main text.

## H.1  UNCONSTRAINED OPTIMIZATION

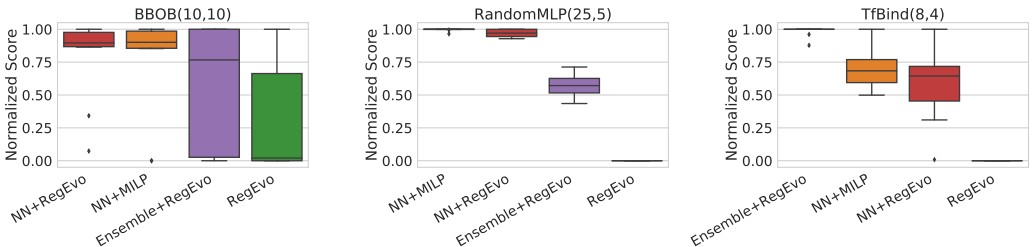

Figure 8: Distribution of algorithms' normalized AUC scores (Section H.1) on unconstrained problems split by objective function class. Higher is better. Relative performance of algorithms in terms of AUC is similar as best-observed reward (Figure 1).

While the best observed reward in $\mathcal{D}_N$ (i.e. after all evaluations) is the primary metric of comparison for algorithms per Section 3.1, it is also instructive to consider a measure of how fast algorithms converge to their best observed reward. To this end, we define an AUC metric that computes the *area under the best observed reward curve*; higher values indicate that an algorithm found better points in earlier iterations. To faciliate comparison across problems, we min/max normalize algorithms' AUC scores within each problem exactly as we did for the best observed reward (Section 4.1). That is, the best (resp. worst) on-average algorithm in terms of AUC is assigned a score of one (resp. zero) and intermediate values express relative distance from these extremes.

Figure 8 plots the distribution of algorithms' AUC scores over all unconstrained problems, split by objective function class. The relative performance of algorithms in terms of this new AUC metric does not differ significantly from what we found for final reward (Section 4.2, Figure 1). Figures 11 and 12 plot the individual reward curves as function of outer-loop iteration for each unconstrained problem.

## H.2  CONSTRAINED OPTIMIZATION

In Figure 9 we plot the distribution of algorithms' normalized final reward scores across all constrained problems from Section 4.3 (paralleling Figure 1, top). The individual reward curves for all such problems can be found in Figures 13 and 14, which did not differ significantly.

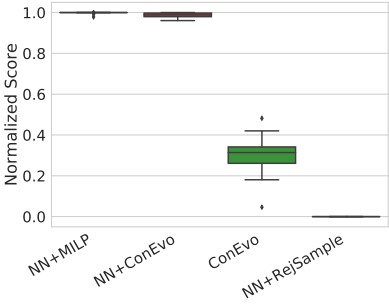

Figure 9: Distribution of algorithms' normalized scores (Section 4.1) on constrained problems. Higher is better. NN+MILP and NN+ConEvo perform similarly across all problems. Reward progression on individual problems can be found in Figure 13 and 14.

### H.3 PRACTICALITY OF MILP

Table 2: Distribution of per-step MILP inner-optimization solve times in seconds for TfBind8 benchmarks when using different surrogate network architectures. The *Network* column denotes the number of ReLUs in each fully-connected hidden layer. Runs were given a time limit (TL) of 300s. (*) means that the time limit was hit.

| Network | min | med | 95% | 99% | max | %TL |
|---|---|---|---|---|---|---|
| Linear | 0.004 | 0.4 | 1.4 | 2.9 | 16.9 | 0% |
| FC(16) | 0.02 | 2.2 | 8.0 | 15.5 | 60.8 | 0% |
| FC(32) | 0.04 | 11.7 | 49.2 | 85.5 | 300* | 0.1% |
| FC(16,16) | 0.40 | 12.2 | 55.6 | 109.1 | 300* | 2.1% |

Here we present experiments to explore the impact of surrogate network size on MILP solve times. We varied *NN+MILP*'s surrogate network architecture using fully-connected networks with different numbers of hidden layers and neurons, FC(16), FC(32), FC(16,16), as well as a simple Linear model (no hidden layer). Note that the first architecture, FC(16), was what was presented in the main results. Otherwise, we used the same training and optimization hyper-parameters for *NN+MILP* as described in Section C.1. We ran this experiment on all 12 unconstrained *TfBind* problems from Section 4.2, with 20 trials using different random initial data sets.

Table 2 shows aggregate distribution statistics of inner-loop optimization runtime for the different architectures, across all steps of all trials of all problems. We note that solve times increase as the network size increases, but even for the largest network (two layers with 16 neurons each), the solver rarely times out and almost always terminates within a practical time limit. Furthermore, for larger networks we can improve scaling using advanced formulation techniques (e.g. Appendix A). We also note that, in these experiments, there was no single architecture that consistently produced better optimization across different instances (though *Linear* was almost always outperformed by the rest).

We also include Figure 10 which plots the distribution of MILP inner-loop solver runtimes for all constrained and unconstrained problems as a function of outer-loop iteration (paralleling Figure 2b which included only TfBind). As noted in Section 4.4, the relationship between problem size and runtime can be unpredictable. For example, the lower-dimensional TfBind problems showed the highest mean and variance in MILP inner-loop solve times among all constrained and unconstrained objective classes. Encouragingly, across all problem classes we observe only a linear (roughly) increase in solve time per iteration, presumably due to the increasing number of no-good constraints.

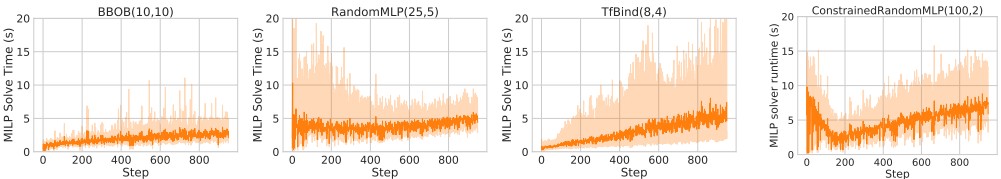

Figure 10: Distribution of MILP acquisition problem solve times as a function of iteration split by objective class for unconstrained problems (Section 4.2) and for all constrained problems (Section 4.3). Line and bands show the median and 5th/95th percentile range over all trials of all problems in a class.

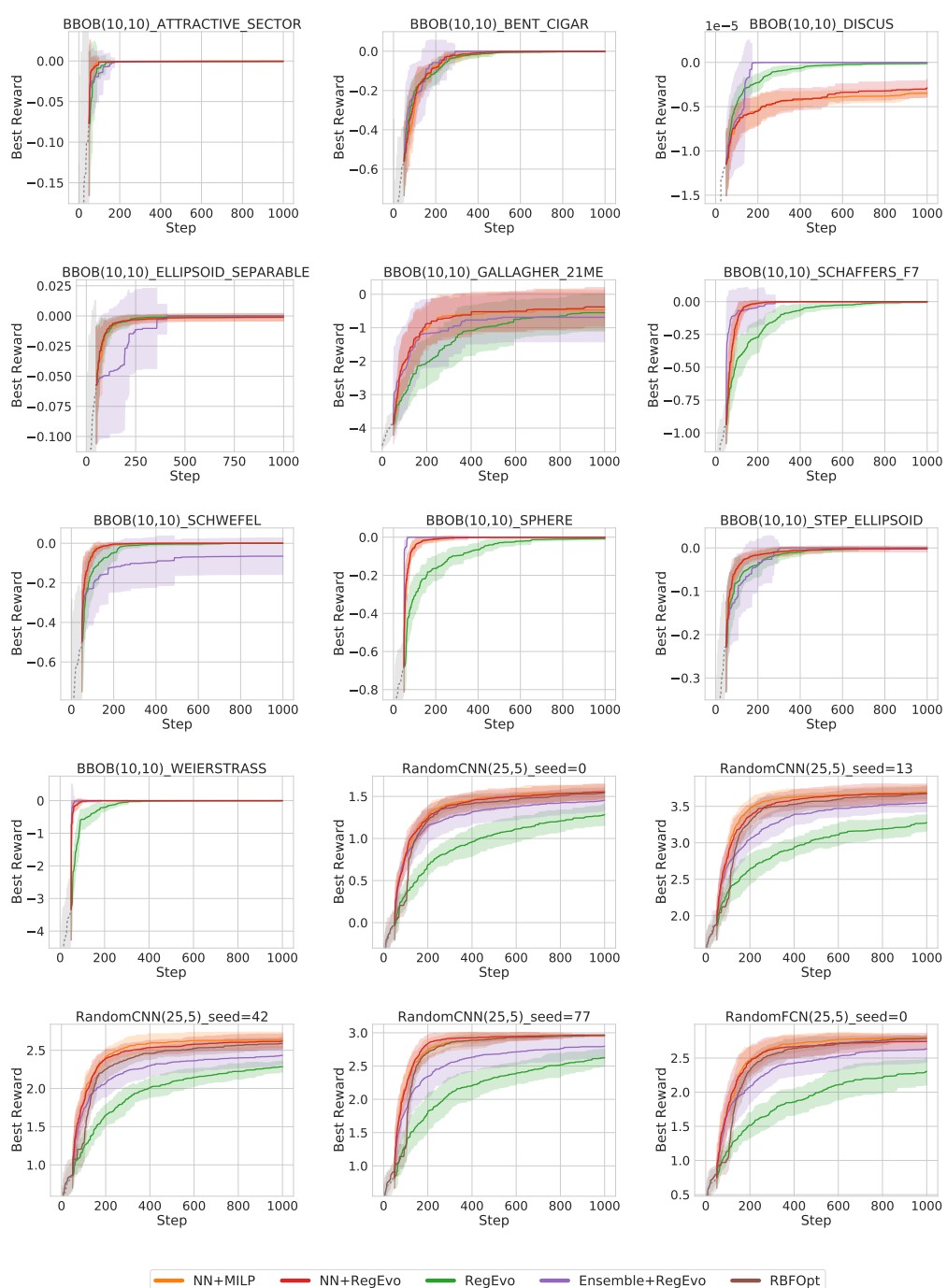

Figure 11: Best observed reward as a function of iteration for the first half of all unconstrained problems (Section 4.2), averaged over 20 iterations (bands indicate ±1sd). Dashed grey lines in the first 50 steps indicate the initial randomly sampled dataset, common to all methods except RBFOpt, which performs its own initialization.

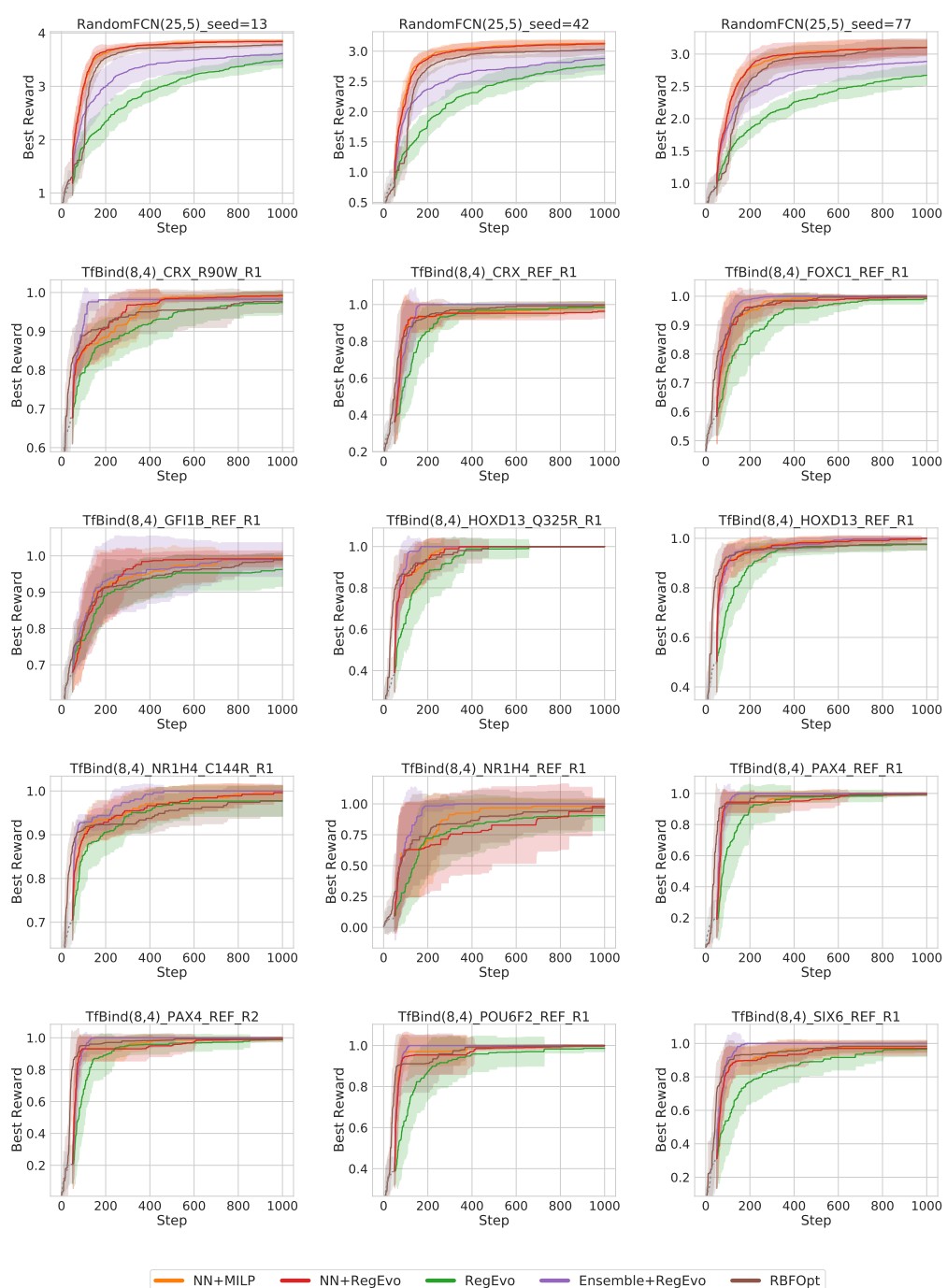

Figure 12: Best observed reward as a function of iteration for the second half of all unconstrained problems (Section 4.2), averaged over 20 iterations (bands indicate ±1sd). Dashed grey lines in the first 50 steps indicate the initial randomly sampled dataset, common to all methods except RBFOpt, which performs its own initialization.

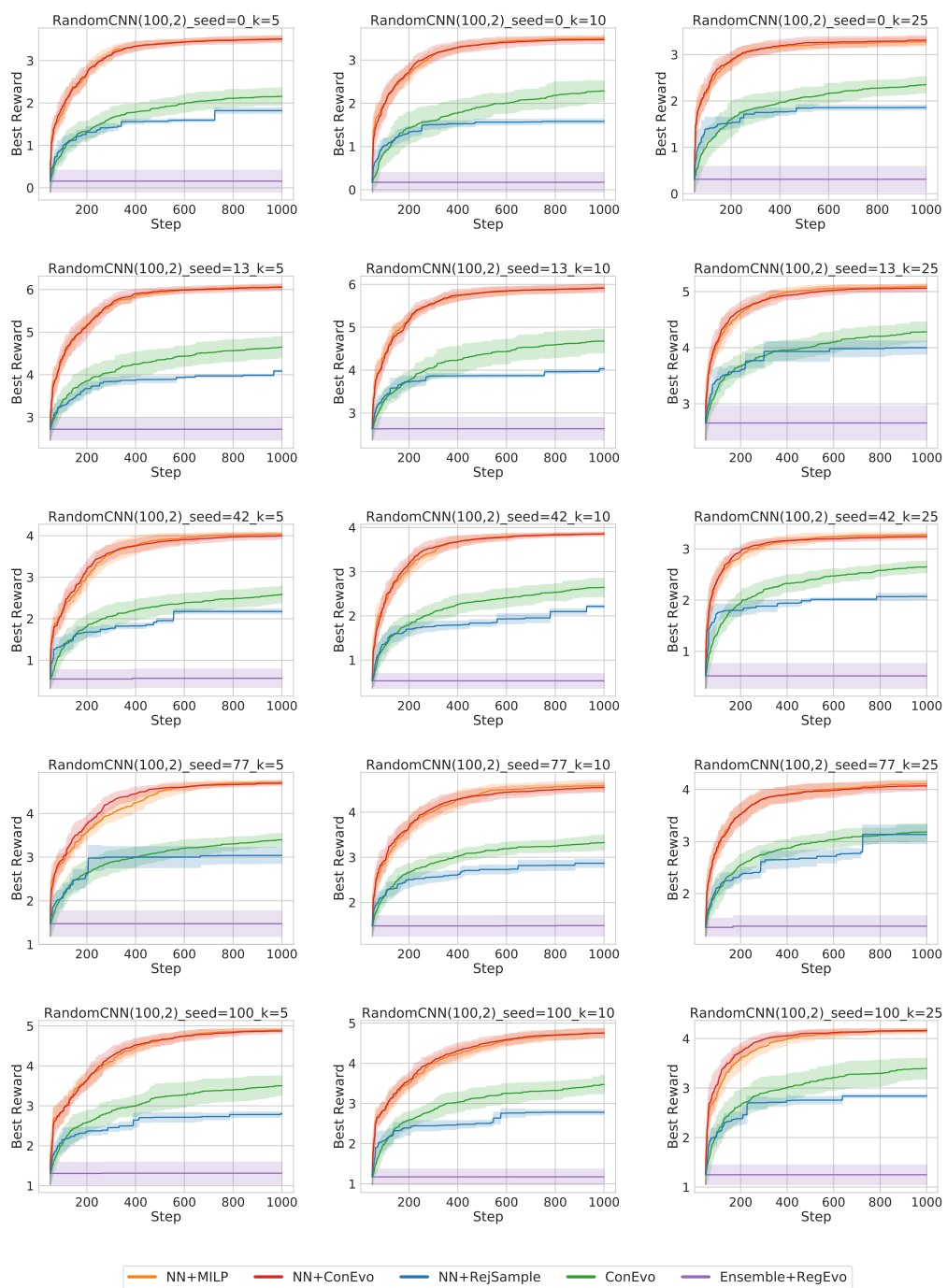

Figure 13: Best observed reward as a function of iteration for the first half of all constrained problems (Section 4.3), averaged over 20 iterations (bands indicate ±1sd). Initial randomly sampled set of 50 points is omitted.

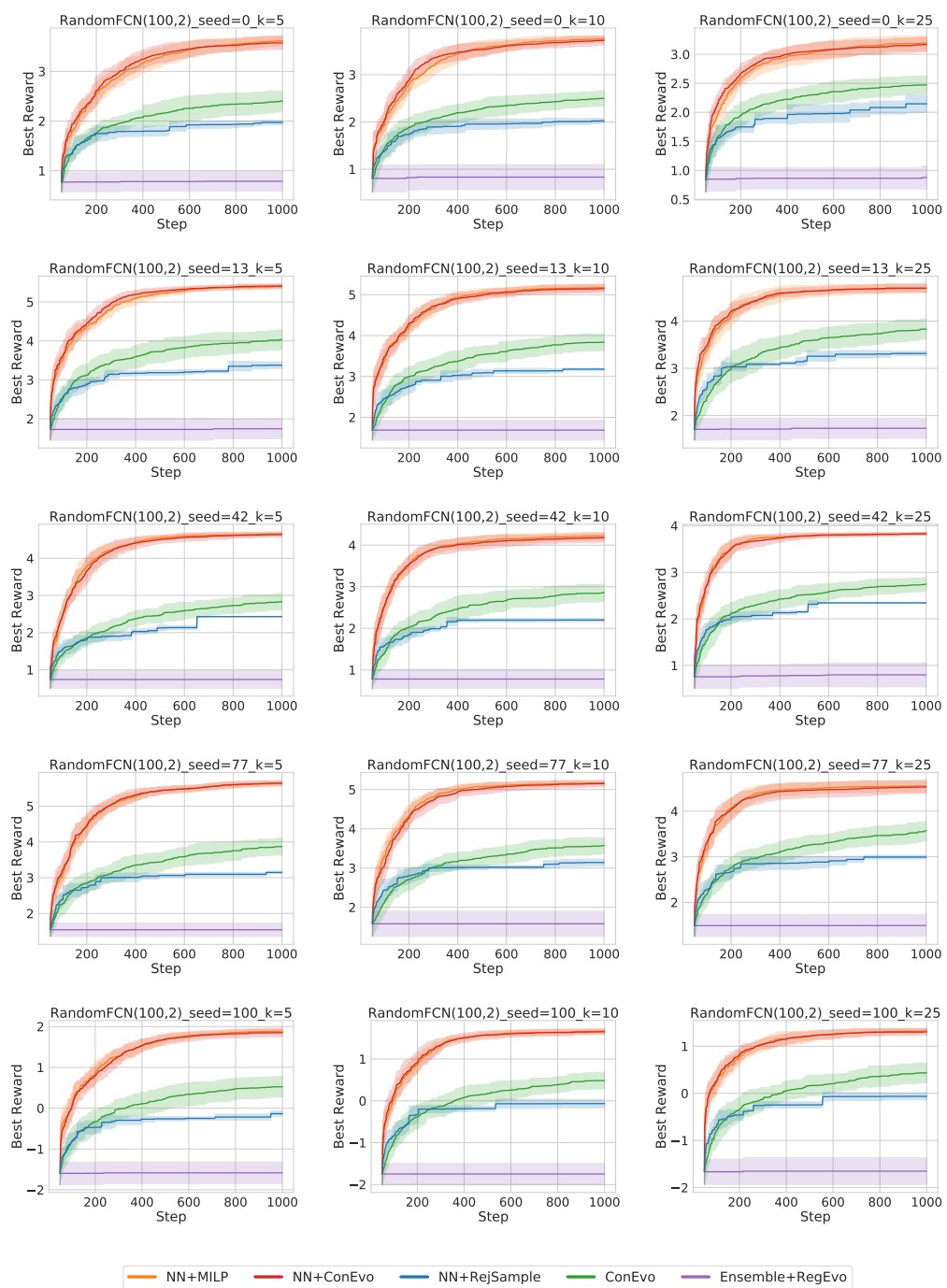

Figure 14: Best observed reward as a function of iteration for the second half of all constrained problems (Section 4.3), averaged over 20 iterations (bands indicate ±1sd). Initial randomly sampled set of 50 points is omitted.

