# OpenReview forum: "Constrained Discrete Black-Box Optimization using Mixed-Integer Programming"
_ICLR.cc/2022/Conference — ICLR 2022 Submitted_

### Official Review · Reviewer_vgXk · 2021-10-31

**Correctness:** 3
**Technical Novelty And Significance:** 3
**Empirical Novelty And Significance:** 4
**Recommendation:** 6
**Confidence:** 4

**Main Review:**

# Strengths
This paper tackles an interesting problem of optimizing expensive blackbox functions even in the case of complex combinatorially constrained domains, by using a MILP formulation of a piecewise linear neural network. Even though such formulations have been used before (the authors mention neural network certification as an example), it offers a very flexible way of easily integrating combinatorial constraints, without handcrafting a local search method to maintain feasibility of solutions to the acquisition problem. The experiments show, that
1) solving the acquisition problem to global optimality can provide benefits over local search methods even in case of unconstrained domains
2) performance of handcrafted methods to ensure proposal feasibility in the constrained case can either be matched (NN+ConEvo in RandomMLP with subset constraints) or surpassed (RE in NAS-101), with a simpler way to implement the constrained domain in a general framework.
With the flexibility of the framework this can be used for various other future problems that involve MBO in the presence of combinatorially constrained domains.

# Weaknesses
### Limitations of surrogate model
The surrogate model is limited to a) neural networks with piecewise linear activation functions, this is inherent to the MILP formulation and restricts the framework from using potentially better-suited surrogate hypothesis classes such as Random Forests.
The surrogate model is also limited to b) a relatively small number of neurons in the network due to runtime limitations of the MILP. The dimensionality and number of constraints in the MILP formulation scales linearly with the number of nonlinearities in the network and as mentioned in the paper, the runtime scaling of the MILP solver is then often unpredictable.
This means that the runtime comparison between methods is a crucial factor in the comparison, which leads to some questions.

1) In section 4.4 the authors reported the inner-loop optimization runtimes for the unconstrained case. To me it is very surprising that the local evolutionary search RegEvo has a larger average runtime than the global MILP optimization. It would be great if the authors could provide intuition on why this is the case. Additionally, is this also the case when comparing MILP to ConEvo in the constrained case, where those two methods are the main competitors? As the constrained case is the one this framework is designed for, this would be the more important comparison than in the case of an unconstrained domain.
2) In the experiments a single hidden layer with 16 neurons is used, and ablations with two slightly bigger architectures (32 and 16+16 neurons) are provided. While the slightly bigger architectures are still within the computational limits, the network probably cannot be scaled to much larger (deep) networks. In the ablations it is stated that for the TFbind8 experiment larger architectures did not improve performance, which suggests that the small surrogate model is already expressive enough in this case. It would be interesting to know whether this is also the case for a larger experiment such as the NAS case study.

In any case, the runtime restriction could be a limiting factor in future applications, where more complex surrogates could be required, giving faster but less optimal local search methods an advantage over the global optimization.

### Integration of continuous variables
In the current stage the framework doesn’t support mixed-integer domains because the no-good constraints do not naturally extend to this case, as stated in the conclusion. Therefore, in its current stage, the method should be called NN+ILP instead of NN+MILP, as the name otherwise suggests that mixed-integer domains can be used.


## Typos
These do not affect my rating.
1) Before section 2.2: ‘are’ should be removed

**Summary Of The Paper:**

The authors present NN+MILP, a framework for the optimization of an expensive to evaluate blackbox fuction with a discrete combinatorially constrained domain. The acquisition problem of finding the surrogate minimum is solved to global optimality by solving a MILP formulation of the acquisition problem. The MILP formulation limits the considered neural network surrogate class to networks with piecewise linear activation functions. However, it provides a simple declarative language for integrating the problem-specific constraints.

The experiments cover both the cases of unconstrained and constrained optimization of the acquisition function.
The unconstrained case compares NN+MILP to general purpose algorithms for unconstrained discrete blackbox optimization. It shows that the global optimization often achieves better results than a local search evolution based method for solving the acquisition problem, while also solving the inner loop problem faster. Additionally, it demonstrates that even with the restrictive function class of a single layer neural network, comparable performance to better-suited surrogate hypothesis classes can be achieved, thanks to the global optimization.
In the constrained case artificial subset-equality constraints are used to create a combinatorial domain for the blackbox function. NN+MILP performs similar to NN+ConEvo, a manually adjusted local search method that ensures feasibility of the proposals at every step. Other methods employing random search for inner loop optimization or local search with a different surrogate hypothesis class perform much worse.
Finally, a case study for the NAS-Bench-101 benchmark is provided. A novel MILP formulation for cells of a valid architecture design is described, which consists of an MILP formulation for directed acyclic graphs with added null operations to allow DAGs with a reduced number of cells. Despite the generality of NN+MILP it outperforms a strong evolution based baseline.

**Summary Of The Review:**

Strengths: The presented framework is flexible and easier to implement than other methods in the presence of combinatorially constrained discrete domains. The experiments show that the performance is also competitive with other approaches.

Weaknesses: The MILP formulation limits the hypothesis class of models to neural networks with linear activation functions, runtime limitations restrict the number of neurons in the network. Mixed-integer domains are currently not supported by the framework, even though the name suggests otherwise.

---

> ### Author Response · Authors · 2021-11-17
> **Response to Reviewer vgXk**
>
> We thank the reviewer for their detailed comments. We address the concerns below:
>
> **Runtimes of NN+MILP vs NN+RegEvo** Modern MILP solvers are used to solve practical problems with many thousands of variables and constraints in seconds, whereas our acquisition problems only have a few hundred. When using an evolutionary solver in the inner loop (RegEvo or ConEvo), the main driver of runtime is that they have to generate 10,000 new random mutations/recombinations of existing sequences.
>
> We also note that NN+MILP’s runtimes in the constrained RandomMLP experiments are shown in Figure 8 (Appendix H.3). In that case, the mean and standard deviations of the optimization time at each step are 10.87s ± 3.74s for NN+MILP and 43.81s ± 15.88s for NN+ConEvo. Compared to the unconstrained version, there is a larger gap in performance since the mutation operator is more expensive, and the problem has more variables despite the smaller vocabulary size, (whereas with NN+MILP all problems are binarized anyway).
>
> **Scalability of NN+MILP** The main reason why we chose small networks is not performance – we solve the MILPs in a few seconds and could afford to spend more time – although it is of course a factor. Instead, the bottleneck for good approximation is the size of the data we are training on. Here, we are dealing with at most a few hundreds data points and relatively small-dimensional inputs. To further support the network-size ablation tests from TfBind (Appendix H.3), we have performed additional experiments on problems with larger domains (constrained subset selection with n=200 and n=400 items), where we also do not observe substantial improvement when using 32 neurons instead of 16 (please see Figure 6 in the updated Appendix E). That said, we acknowledge that the capacity of the network may affect approximation in large instances. This is suggested to some extent in the early rounds of Figure 6 (right).
>
> **MILP terminology** As stated, the no-good constraints do preclude optimizing over mixed-integer domains, and we acknowledge that the term MILP might cause confusion in this regard. However, a mixed-integer model is necessary for our approach since optimization over a neural network requires continuous variables as the outputs of neurons. We will clarify this distinction in the camera-ready.
>
> We also thank the reviewer for pointing out the typo in Section 2.1, we have corrected it in the updated version.

---

> > ### Comment · Reviewer_vgXk · 2021-11-18
> > **Further Questions**
> >
> > Thank you for answering my questions, this mostly resolved my initial concerns. I agree that for the limited number of sampled data points which the surrogate model is trained on, the size of the surrogate model is large enough.
> >
> > However, I agree with the other reviewers that the missing exploration-exploitation tradeoff should be discussed in more detail. I understand that in the case of a combinatorially constrained discrete domain the focus is not on modeling uncertainty (as it would usually be possible with a GP), and the authors argue that the stochasticity in training the surrogate offers enough randomness to lead to sufficient exploration. But I could imagine that the surrogate will propose points in the region of a local optimum exhaustively until all points in that region are removed with no-good constraints, as the new points in the locally optimal region will probably not dramatically change the fit of the trained surrogate. This exhaustive behavior would not have been necessary if by more randomness a new point would have been sampled that has more impact on training the surrogate at the next iteration. Have the authors considered more explicit exploration, maybe by proposing a random feasible point instead of the surrogate optimum with epsilon probability?

---

> > > ### Author Response · Authors · 2021-11-18
> > > **explicitly encouraging exploration**
> > >
> > > Thanks for the questions regarding exploration! We could  sample from the posterior over neural network weights using, for example, Hamiltonian Monte Carlo or variational inference. Then, our algorithm would correspond exactly to Thompson sampling, which is well-established in the Bayesopt and Bandits literature.
> > >
> > > As you suggest, however, we could explicitly encourage exploration using, for example, an epsilon-greedy approach where occasionally we sample a random point. It's possible that this would help, but we see it as orthogonal to the contributions of the paper and not gating for acceptance. I can also imagine another exploration strategy where we explicitly ensure that the next point we query f(x) with has some minimum hamming distance from other points that we have recently queried. This could be achieved by adding additional linear constraints to the MILP. We aren't aware of a natural way to achieve such constraints using, for example, an evolution solver to optimize the acquisition function.
> > >
> > > Overall, we think that using MILP opens up lots of doors for innovating new Bayesopt approaches, and we think that the ICLR community will benefit from learning this.

---

### Official Review · Reviewer_qsFh · 2021-11-01

**Correctness:** 3
**Technical Novelty And Significance:** 2
**Empirical Novelty And Significance:** 2
**Recommendation:** 3
**Confidence:** 4

**Main Review:**

### Reasons to Accept

+ This paper is well-written and well-organized.
+ It solves a very interesting problem that contains constraints on discrete search space.

### Reasons to Reject

- I do not think that a piecewise neural network models an uncertainty of unknown objective appropriately.
- Following the point described above, even though we train a neural network every iteration, it tends not to reflect a factor for exploration, which implies that regression results are almost same where the same observations are given.
- GP is a popular choice for a surrogate function, since it has sufficient expressiveness, which is defined on a RKHS space. I am curious that the surrogate function used in this paper is sufficiently expressive of unknown function.

### Questions to Authors

Please answer the comments described in Reasons to Accept and Reasons to Reject.

1. Can I ask what the difference between no-good constraints and generic equality constraints is? If they are similar, it can be considered in a continuous search space?
1. Is there any specific reason not to compare the proposed method to COMBO? In my experience, COMBO can be thought of as the state-of-the-art model now. To cope with a no-good constraint, you can just apply a rejection strategy in a step of acquisition function optimization (i.e., local search with rejection sampling).

**Summary Of The Paper:**

This paper solves a constrained discrete black-box optimization problem that employs a surrogate model in modeling an unknown objective function. Unlike the formulation of standard Bayesian optimization, it constructs a surrogate model using a piecewise linear neural network. Under the assumption that a randomly-initialized neural network is able to produce an uncertainty for exploration (against exploitation), the proposed method optimizes a surrogate model directly with mixed-integer linear programming. It follows a spirit of Thompson sampling. Finally the authors conduct their method on several experimental circumstances and show the validity of their method.

**Summary Of The Review:**

It solves a very interesting topic which is defined on a discrete search space with constraints. However, the choice of surrogate model is not convincing and the baseline is missing. Thus, I would like to recommend rejection.

---

> ### Author Response · Authors · 2021-11-17
> **Response to Reviewer qsFh**
>
> We thank the reviewer for their detailed comments. We address the concerns below:
>
> **Exploration in NN+MILP** Exploration is encouraged in NN+MILP through a combination of stochasticity in training and no-good constraints. Surrogates are trained from scratch at every iteration, with significant randomness (random re-initialization of weights, stochastic GD, bootstrapping of the training set), which should result in trained networks with variance in their local maxima. Even if the maxima coincide from iteration to iteration (plausible once a local area has been sampled several times), the no-good constraints of Section 3.3 will explicitly prevent the inner-loop optimizer from proposing these points again, and would guide NN+MILP to less-explored areas. Informally, our method performs Thompson sampling. If we used Hamiltonian Monte Carlo instead of SGD to sample from the posterior over parameters, it would be exactly Thompson sampling. We will update the main text of the paper to make this more clear.
>
> **Expressivity of neural networks** Like GPs, shallow neural networks offer universal approximation guarantees. We have considerable practical experience using both kinds of models and have found that both can perform well on datasets with the small scale seen in this paper. The key differentiator between the models is not their expressivity, but how well they enable practical acquisition function optimization, particularly for constrained domains.
>
> **No-good constraints for continuous domains** The approach does not directly generalize to continuous domains since solutions can lie in the interior of the feasible set, and thus would result in nonconvex constraints which are much more difficult to handle.
>
> **Using COMBO with rejection sampling** Given the performance of NN+RejSample, we do not believe that rejection-sampling based search would be practical for general constrained domains. For example, in the graphpart_clique instances in Appendix F, the constraints are partitioning constraints in which we are given several triples of variables and we need to choose exactly one element out of each of them. Letting $t$ be the number of triples, the number of feasible points is $3^t$ whereas the total number of binary points is $2^{3t}$. Even for the smallest instance with 20 triples, the ratio between feasible solutions and number of points is approximately $3 \times 10^{-9}$, which makes it impractical for rejection sampling. That said, one could embed COMBO with a constraint-specific sampling algorithm, but one of the main benefits of our approach is its generality in handling constraints, including those where feasibility is NP-complete.

---

> > ### Comment · Reviewer_qsFh · 2021-11-23
> > **Response to Authors**
> >
> > Thank you for your kind response.
> >
> > Most of my concerns are resolved, but I would like to ask more about the exploration in NN+MILP.
> >
> > I realized there exist many randomness factors, which would help to explore an unexplored region, but I am not sure they are explained an uncertainty we would like to know well.
> >
> > I am not an expert of uncertainty quantification fields, but aleatoric uncertainty can be easily estimated while it is difficult to estimate epistemic uncertainty.
> >
> > Could I ask if the randomness factors you mentioned, e.g., random initializations of weights, SGD, bootstrapping of the training set, and no-good constraints, can induce the well-explained epistemic uncertainty?
> >
> > Additionally, please compare it to GP regression, which generally works well in uncertainty quantification.

---

> > > ### Author Response · Authors · 2021-11-23
> > > **epistemic vs. aleatoric uncertainty**
> > >
> > > Yes, our method is capturing epistemic uncertainty, since it is maintaining uncertainty over the weights of the neural network regressor.
> > >
> > > Consider a regression model P(y | x, theta), where we seek to predict y given input x and model weights theta. Aleatoric uncertainty describes the noisiness in this distribution over y. It may be heteroskedastic, in that the shape of the distribution (e.g., the variance) depends on x. Epistemic uncertainty describes our uncertainty about theta, given a dataset of {x, y} pairs.
> > >
> > > Our method can work with any approach for (approximately) sampling from the posterior distribution over theta, given a training set. Sampling from the posterior captures our epistemic uncertainty. We use a popular heuristic: training a neural network from scratch with random initialization [1]. However, we could have easily used alternative posterior sampling methods, such as MCMC or variational inference. See [2] for an overview. How we do the sampling is orthogonal to the contributions of our paper.
> > >
> > > Like our approach, GP regression also captures epistemic uncertainty. The goal of our paper is not to argue that neural networks are better than GPs for Bayesian optimization. There are many modern papers that have considered both kinds of regressors (e.g., [3]). Instead, our focus was on the impact of exact vs. inexact optimization of the acquistion function, particularly for combinatorially-constrained problems.
> > >
> > > I also disagree that aleatoric uncertainty is easier to estimate that epistemic uncertainty. Note that epistemic uncertainty decreases with the amount of training data, whereas aleatoric uncertainty is 'unexplainable.'
> > >
> > > [1] https://arxiv.org/pdf/1912.02757.pdf
> > > [2] https://nips.cc/media/neurips-2020/Slides/16649.pdf
> > > [3] https://arxiv.org/pdf/1502.05700.pdf

---

> > > > ### Comment · Reviewer_qsFh · 2021-11-24
> > > > **Response to Authors**
> > > >
> > > > Right, I agree that your method does not need to outperform GP regression in uncertainty quantification,
> > > >
> > > > And also, the discussion on this topic of uncertainty quantification might be needed in the paper, but if you think this discussion disturbs the main point of this paper, it might be able to be omitted.
> > > >
> > > > > I also disagree that aleatoric uncertainty is easier to estimate that epistemic uncertainty. Note that epistemic uncertainty decreases with the amount of training data, whereas aleatoric uncertainty is 'unexplainable.'
> > > >
> > > > It is my mistake. Right, we cannot say that capturing aleatoric uncertainty is easy, but a model tends to capture it where a sufficient amount of data is given.

---

### Official Review · Reviewer_j85k · 2021-11-02

**Correctness:** 3
**Technical Novelty And Significance:** 3
**Empirical Novelty And Significance:** 3
**Recommendation:** 6
**Confidence:** 4

**Main Review:**

Overall, the paper is well-written and suggests an interesting and relevant approach to address black-box optimization problems equipped with discrete domains. I found the basic framework to be well thought-out and, in my view, of potential value for a large array of settings within this field.

My major concern, however, is that many design choices are somewhat unclear, and the paper often feels that lacks depth in more specific areas. In particular:

(1) Optimizing inputs with neural networks is challenging with MILPs, often requiring more sophisticated implementations as that of Anderson et al., 2020 (as nicely emphasized by the authors). I wonder whether the fact that the authors were limited to a very simple network, with only a single layer, did not hinder some key insights on the numerical evaluation? For instance, wouldn't the function approximation be quite poor for larger instances? Also, is this approach really scalable?

Perhaps my suggestion here is for authors to consider encodings that are much more efficient/scalable for MILPs (and other model-based approaches). For example, many black-box functions have a natural discrete structure, and authors could consider a decision tree, which is much "simpler" to optimize.

Similar reasoning applies to the "no-good" constraints, which in this case are quite simple and only eliminate one solution at a time. This could really impact MILP performance (e.g., similar to combinatorial Benders cuts). Would there be cases where it is possible to eliminate several points from the acquisition domain? For example, suppose "x" are binary and represent subsets. If you eventually learn all subsets of size <= N, you could eventually replace those constraints by \sum_i x_i >= N + 1.

(2) The numerical experiments do not seem to reflect well the benefits of the approach. To the best of my understanding, the final conclusion is that problems are "easier" to model with NN+MILP, but performance improvements are marginal (if any). I believe this is indeed the case, but the paper lacks more concrete evidence of this statement.

In particular, when comparing NN+MILP and NN+ConEvo, why not consider a problem class that the MILP has clear benefits? For example, a set-packing/set-covering acquisition domain, perhaps with other side constraints, or a scheduling feasible set (e.g., "f" could encode a weighted completion time with unknown weights, and the feasible set are the valid sequences). The authors could compare with any global optimizer (as opposed to RejSample and ConEvo)  because it is well-known that MILP is one of the state-of-the-art techniques for these problems.

**Summary Of The Paper:**

The paper develops a technique to optimize an unknown, black-box function "f" by leveraging a combination of neural networks with mixed-integer linear programming (MILP) methodology. More specifically, authors encode an approximation of "f" using a neural network with piecewise-linear activation functions, which is optimized using its associated reformulation as an MILP with no-good cuts. Numerical results evaluate the approach with respect to other baselines and neural network optimization mechanisms.

**Summary Of The Review:**

The ideas are novel and significant, especially given the modeling expressivity provided by MILPs. However, the paper lacks some justification concerning the scalability of the approach and the fact that neural networks had quite a limited size. Moreover, in my view, the numerical experiments do not explore well the benefits of their approach.

---

> ### Author Response · Authors · 2021-11-17
> **Response to Reviewer j85k**
>
> We thank the reviewer for their detailed comments. We address the concerns below:
>
> **Empirical improvements in constrained setting** We have updated the manuscript with experiments on constrained problems with larger domains, where we observe that solving the inner problem to optimality can indeed be beneficial (please see Figure 5 in the updated Appendix E). The experimental setup is identical to Section 4.2, but with an increased number of items (n=200 and n=400) in the subset selection problem. We observe that NN+MILP significantly outperforms NN+ConEvo in terms of maximum reward for the largest set of instances (n=400). The difference is less pronounced for n=200, but NN+MILP does exhibit higher rewards in early rounds.
>
> Regarding other settings where MILP is expected to have clear benefits, we agree and indeed Appendix F already contains experiments very similar to what the reviewer suggests: binary quadratic problems from MINLPLib, where there is no natural evolutionary baseline. We chose these because QPs are possible to optimize via other methods and there are good feasible solutions to compare with, some of which are known to be optimal. We do not compare directly against QP solvers in solve time because they exploit white-box access to the objective function, but we do observe that we can match the best known value provided by MINLPLib in many cases despite using a black-box solver that knows nothing about the QP structure.
>
> **Scaling to larger surrogate networks** The main reason why we chose small networks is not performance – we solve the MILPs in a few seconds and could afford to spend more time – although it is of course a factor. Instead, the bottleneck for good approximation is the size of the data we are training on. Here, we are dealing with at most a few hundreds data points and relatively small-dimensional inputs. To further support the network-size ablation tests from TfBind (Appendix H.3), we have performed additional experiments on problems with larger domains (constrained subset selection with n=200 and n=400 items), where we also do not observe substantial improvement when using 32 neurons instead of 16 (please see Figure 6 in the updated Appendix E). That said, we acknowledge that the capacity of the network may affect approximation in large instances. This is suggested to some extent in the early rounds of Figure 6 (right).
>
> **Improved no-good constraints** We agree that stronger no-good constraints might prove useful for scaling the method up further, and we have considered them. We opted not to include them here because solve times were already only a few seconds, and as can be seen in Figure 10 (Appendix H.3), the no-good constraints do not contribute significantly to a degradation in solve time in our experiments. We will consider adding a discussion on this as a possible improvement in the Appendix.

---

> > ### Comment · Reviewer_j85k · 2021-11-22
> > **Acknowledgement**
> >
> > Thank you for the reply, I appreciate it and acknowledge your response. I believe you addressed the majority of my concerns and I believe the new experiments add important insights. I wonder how those comments would be more clearly presented in the main text, as opposed to only the Appendix?

---

> > > ### Author Response · Authors · 2021-11-22
> > > **Response to Reviewer j85k**
> > >
> > > Thanks for the response. We would indeed like to highlight the new experiments in the main text for the camera-ready version and we are flexible with rearranging the paper. For example, we could:
> > >
> > > 1. Expand on the main topics discussed in this review, including with other reviewers, such as scalability and exploration vs exploitation. We may need to readjust some of the other text to make this fit. We can also extend the appendix with a few of the other minor points raised in this review, such as strengthening the no-good constraints.
> > >
> > > 2. Move the new experiments to the main text. This would require us to replace one of the plots, however, by the experiment of size 400 (any plot we replace would be kept in the appendix). Two possible candidates for replacement would be either Figure 2(b) (the solve times) or the bottom row of Figure 1 (the examples for the unconstrained case). Perhaps if we remove the larger figure (bottom row of Figure 1) we might have space to discuss some of the MINLPLib experiments in the main text as well; we will think about this carefully.
> > >
> > > This is tentative and we would be happy to take suggestions.

---

### Official Review · Reviewer_5edm · 2021-11-04

**Correctness:** 4
**Technical Novelty And Significance:** 2
**Empirical Novelty And Significance:** 2
**Recommendation:** 5
**Confidence:** 4

**Main Review:**

### Strong points
- The declarative nature of the proposed method eliminates the need for developing an algorithm for solving the inner loop optimization (as long as the structural constraints can be represented in MIP formalsim). It also facilitates to adpating an existing method to similar variations by simply modifying the constraints.

### Aspects to be improved
- The choice of a piecewise linear neural network for both the surrogate model and acquisition function is not properly motivated. In particular, it is not clear how the proposed approach maintains an exploration-exploitation balance.

- The proposed approach does not demonstrate significant improvements in empirical evaluations. In the unconstrained setting, it is outperformed by the baseline, and in the constrained setting, solving the inner loop problem to optimality does not seem to provide an advantage. The argument of "ease of implementation" would have been convincing if implementing the alternative (evolutionary) approach was prohibitively difficult. But this does not seem to be the case.

- The proposed approach relies on solving a MIP problem repeatedly. This does not allow the method to be applied to problems with more than a certain number of variables. The problems studied in the  experiments have few number of variables. In particular, *TfBind(8, 4)* can be probably solved by simple enumeration. It is not clear to what extent the proposed method is applicable to larger problems.

### Question

- When comparing different methods, each algorithm is evaluated in terms of the best reward observed after 1000 queries. At each iteration, the MIP solver is given a 500 seconds timeout, and the evolutionary aglorithms is given a budget of 10k queries. Does this give the inner loop algorithms equal opportunities for finding good solutions? Isn't it more fair to compare the methods subject to an equal overall time budget?


**Summary Of The Paper:**

The paper proposes a method using piecewise-linear neural networks as the surrogate model (and the acquisition function) in a black-box optimization framework. At each step, the acquisition function is optimized by casting the learned neural network (and a set of constraints excluding the already-visited data points) into a mixed integer linear program, which can be then solved using off-the-shelf solver. The proposed method is empirically evaluated on a number of unconstrained and constrained tasks.










**Summary Of The Review:**

The paper presents an interesting direction for using a declarative framework (i.e. mixed integer linear programming) to encode combinatorial structures in black-box optimization. The proposed method lacks a theoretical motivation, and the empirical evaluation does not demonstrate significant advantages over existing methods.

---

> ### Author Response · Authors · 2021-11-17
> **Response to Reviewer 5edm**
>
> We thank the reviewer for their detailed comments. We address the concerns below:
>
> **Empirical improvements in the constrained case** We have updated the manuscript with experiments on constrained problems with larger domains, where we observe that solving the inner problem to optimality can indeed be beneficial (please see Figure 5 in the updated Appendix E). The constraint structure is identical to Section 4.2, but with an increased number of items (n=200 and n=400) in the subset selection problem. We observe that NN+MILP significantly outperforms NN+ConEvo in terms of maximum reward for the largest set of instances (n=400). The difference is less pronounced for n=200, but NN+MILP does exhibit higher rewards in early rounds.
>
> **Ease of implementation** We would like to note that we intended NN+ConEvo as an ablation, to show that heuristic optimization approaches can be reasonable (and to the best of our knowledge, in this setting NN+ConEvo has not been previously investigated). However, we do believe that MILP’s declarative nature is significantly more general, since implementing a custom evolutionary approach could be very difficult in many applications. There are many problems where even feasibility is NP-complete, e.g. routing and scheduling problems with time window constraints. In addition, the ability to solve optimization problems with only modeling effort – and zero algorithmic effort – is valuable especially in industry. For (linear) discrete optimization problems, MILP is often the first and final choice as it requires only a model and it is in many cases good enough out-of-the-box, and we believe our work is a contribution towards achieving the same if we replace the linear objective function by a black-box one. We in fact take advantage of this in Appendix F in our paper, where we experiment with a set of binary quadratic problems with a variety of constraints without any modification, whereas running an evolutionary solver would require considerable effort in customizing for each individual type of constraint.
>
> **Evolution should have the same time limit as NN+MILP** As noted in Section 4.4, MILP rarely exhausts the time limit and in fact only takes 7.92s to solve on average, less than the 9s for RegEvo. The time limit of 500s is only intended as a safeguard due to the relatively large variance in MILP solve time. We will highlight this in the camera-ready version.
>
> **Scalability of NN+MILP**  In practice modern MILP solvers scale very well and often solve problems with thousands of variables and constraints to global optimality in minutes, whereas our acquisition problems only have a few hundred. For example, the constrained problems in Section 4.2 have 100 variables in the domain, and solve times are similar to the unconstrained case (Appendix H.3). Moreover, in Appendix F, NN+MINLP finds the best known solution given by MINLPLib for several binary quadratic problems with hundreds of variables. Finally, we note that, though they were not necessary for us given already fast solve times (Section 4.4), we believe that the method can be readily scaled to much larger problems, e.g. by using state-of-the-art commercial MILP solvers (e.g., Gurobi) and using more advanced formulation techniques for larger networks (Appendix A).
>
> **Exploration-exploitation** Exploration is encouraged in NN+MILP through a combination of stochasticity in training and no-good constraints. Surrogates are trained from scratch at every iteration, with significant randomness (random re-initialization of weights, stochastic GD, bootstrapping of the training set), which should result in trained networks with variance in their local maxima. Even if the maxima coincide from iteration to iteration (plausible once a local area has been sampled several times), the no-good constraints of Section 3.3 will explicitly prevent the inner-loop optimizer from proposing these points again, and would guide NN+MILP to less-explored areas. Informally, our method performs Thompson sampling. If we used Hamiltonian Monte Carlo instead of SGD to sample from the posterior over parameters, it would be exactly Thompson sampling. We will update the main text of the paper to make this more clear.

---

> > ### Comment · Reviewer_5edm · 2021-11-22
> > **I acknowledge that I have read the rebuttal**
> >
> > I thank the authors for their detailed response, updating the manuscript, and performing additional experiments. I acknowledge that I have taken their rebuttal, and their responses to other reviewers,  into account in my evaluation.

---

### Author Response · Authors · 2021-11-17

We thank all of the reviewers for their comments, and appreciate the feedback. We address comments in detailed responses to each reviewer. At a high-level, we:

1. Include a new experiment on larger constrained instances (Appendix E) that might alleviate concerns about scalability of NN+MILP [Revs. 5edm, j85k, vgXk] and its performance relative to NN+ConEvo [Revs. 5edm, j85k].
2. Discuss the question of exploration/exploitation balance [Revs. 5edm, qsFh] and suitability of neural network surrogate functions [Revs. j85k, qsFh].
3. Discuss the generality of MILP in handling combinatorial constraints [Revs. 5edm, qsFh].
4. Address other concerns specific to individual reviewers.

---

### Decision · Program_Chairs · 2022-01-20

**Decision:**

Reject

**Comment:**

The paper considers the problem of black-box optimization and proposes a discrete MBO framework using piecewise-linear neural networks as surrogate models and mixed-integer linear programming. The reviewers generally agree that the paper suggests an interesting approach but they also raised several concerns in their initial reviews. The response from the authors addressed a number of these concerns, for instance regarding scalability and expressivity of the model. However, some of these concerns remained after the discussion period, including doubts about the usefulness for typical applications in discrete black-box optimization and some concerns about the balance between exploration and exploration.

Overall the paper falls below the acceptance bar for now but the direction taken by the authors has some potential. I encourage the authors to address the problems discussed in the reviews before resubmitting.